# Multiple lineages of *Streptomyces* produce antimicrobials within passalid beetle galleries across eastern North America

Rita de Cassia Pessotti[1], Bridget L Hansen[1], Jewel N Reaso[1], Javier A Ceja-Navarro[2,3], Laila El-Hifnawi[4], Eoin L Brodie[5,6], Matthew F Traxler[1]*

[1]Department of Plant and Microbial Biology, University of California, Berkeley, Berkeley, United States; [2]Bioengineering and Biomedical Sciences Department, Biological Systems and Engineering Division, Lawrence Berkeley National Laboratory, Berkeley, United States; [3]Institute for Biodiversity Science and Sustainability, California Academy of Sciences, Berkeley, United States; [4]Department of Molecular and Cellular Biology, University of California, Berkeley, Berkeley, United States; [5]Ecology Department, Earth and Environmental Sciences, Lawrence Berkeley National Laboratory, Berkeley, United States; [6]Department of Environmental Science, Policy and Management, University of California, Berkeley, Berkeley, United States

*For correspondence:
mtrax@berkeley.edu

Competing interests: The authors declare that no competing interests exist.

**Abstract** Some insects form symbioses in which actinomycetes provide defense against pathogens by making antimicrobials. The range of chemical strategies employed across these associations, and how these strategies relate to insect lifestyle, remains underexplored. We assessed subsocial passalid beetles of the species *Odontotaenius disjunctus*, and their frass (fecal material), which is an important food resource within their galleries, as a model insect/actinomycete system. Through chemical and phylogenetic analyses, we found that *O. disjunctus* frass collected across eastern North America harbored multiple lineages of *Streptomyces* and diverse antimicrobials. Metabolites detected in frass displayed synergistic and antagonistic inhibition of a fungal entomopathogen, *Metarhizium anisopliae*, and multiple streptomycete isolates inhibited this pathogen when co-cultivated directly in frass. These findings support a model in which the lifestyle of *O. disjunctus* accommodates multiple *Streptomyces* lineages in their frass, resulting in a rich repertoire of antimicrobials that likely insulates their galleries against pathogenic invasion.

## Introduction

The majority of clinically used antibiotics continue to be based on chemical scaffolds derived from natural products (also known as specialized metabolites) made by microbes, namely actinomycete bacteria and filamentous fungi (*Chevrette and Currie, 2019*; *Gholami-Shabani et al., 2019*; *Hutchings et al., 2019*; *Lyu et al., 2020*). However, the spread of resistance among pathogens has led to a steep, and well-documented, erosion in antibiotic efficacy (*Colavecchio et al., 2017*; *Lekshmi et al., 2017*; *Richardson, 2017*). The rapidity of resistance evolution in the medical arena raises questions about how the microbes that make antibiotics preserve their advantageous use over evolutionary time and underscores a need to understand the chemical ecology of microbially produced specialized metabolites.

Symbiotic systems in which actinomycete-derived specialized metabolites are used for chemical defense may provide a blueprint for effectively leveraging antibiotics over long-term timescales. Important examples of such systems include the symbiotic relationships between insects and actinomycetes, in which the insects associate with actinomycetes to protect their food sources, communal nests, or developing larva against pathogenic invasion (*Bratburd et al., 2020*; *Chevrette et al., 2019*; *Li et al., 2018*; *Van Arnam et al., 2018*). Among the most extensively characterized of these systems are the eusocial, neotropical leafcutter ants, who cultivate a food fungus on leaf tissue in their subterranean nests. These ants protect their fungal gardens from a pathogenic fungus (*Escovopsis* sp.) by associating with actinomycetes usually belonging to the genus *Pseudonocardia*, which produce a variety of antifungal molecules (e.g., dentigerumycin and gerumycins) that differentially inhibit the growth of the *Escovopsis* sp. (*Currie et al., 2003*; *Li et al., 2018*; *Menegatti et al., 2021*; *Oh et al., 2009*; *Sit et al., 2015*; *Van Arnam et al., 2016*). Similarly, the gregarious southern pine beetle, which cultivates fungi to feed its larvae, maintains *Streptomyces* sp. capable of inhibiting fungal pathogens (*Scott et al., 2008*). Another archetypal insect/actinomycete system includes the solitary beewolf wasps, which harbor *Streptomyces philanthi* in specialized antennal reservoirs (*Kaltenpoth et al., 2005*; *Kaltenpoth et al., 2010*; *Kaltenpoth et al., 2012*). Female beewolves inoculate their brood chambers with these symbionts, which are ultimately incorporated into the cocoons of their pupating larvae (*Kaltenpoth et al., 2005*). These *Streptomyces* produce a suite of antifungal molecules (piericidins and derivatives, streptochlorins, and nigericin) that protect the brood from opportunistic fungal pathogens (*Engl et al., 2018*; *Kroiss et al., 2010*).

The exploration of these insect/actinomycete associations has provided key insights into the ecology of microbial specialized metabolites. Importantly, analyses of the leafcutter ant and beewolf systems have shown the co-evolution of the insect hosts and actinomycete symbionts, suggesting that these relationships, and the molecules involved, have remained durable over tens of millions of years (*Kaltenpoth et al., 2014*; *Li et al., 2018*). Both leafcutter ants and beewolves have specialized structures for maintaining their actinomycete symbionts, which facilitate vertical transmission and high symbiont fidelity (*Kaltenpoth et al., 2005*; *Li et al., 2018*; *Stubbendieck et al., 2019*). However, outstanding questions remain regarding the nature of actinomycete symbioses in other insects, for example, those without specialized compartments for maintaining bacterial symbionts. Specifically, how do different mechanisms of microbial transmission influence symbiont specificity and diversity? And, what are the implications of symbiont specificity/diversity for the chemical repertoires found in these systems? Thus, we were motivated to identify an insect/actinomycete association that (i) utilized a different mechanism of microbial transmission and (ii) enabled direct detection of microbially produced specialized metabolites in situ.

With this in mind, we assessed *Odontotaenius disjunctus*, a subsocial passalid beetle commonly found in decomposing logs across eastern North America, and its frass (fecal material), as a model system for studying the ecology of actinomycete specialized metabolism. Frass is an abundant and easily sampled material in *O. disjunctus* galleries, and it is an important nutrient source in this system for both adult and larval survival, and pupal chamber construction (*Biedermann, 2020*; *Mason and Odum, 1969*; *Schuster and Schuster, 1985*; *Valenzuela-González, 1992*). Notably, *O. disjunctus* does not appear to have mycangia or other specialized structures that harbor microbial symbionts (*Ulyshen, 2018*). While *O. disjunctus* has not been investigated for the presence of actinomycetes and antimicrobials, a previous study on tropical passalid beetles found a diverse community of actinomycetes inhabiting the gut of both adults and larvae (*Vargas-Asensio et al., 2014*).

We characterized this system through a combination of (i) direct chemical analyses of microbial specialized metabolites in frass sampled from *O. disjunctus* galleries across its geographic range, (ii) parallel assessment of the phylogeny and specialized metabolite repertoire of actinomycete strains isolated from frass, (iii) investigation of synergism/antagonism between the specialized metabolites found in frass against a beetle pathogen, and (iv) direct assessment of competitive interactions between key *Streptomyces* isolates and entomopathogenic strains in an in vitro frass experimental system. Collectively, our results indicate that *O. disjunctus* establish stable associations with a comparatively diverse set of actinomycetes relative to other insect/actinomycete associations, which we propose to be a result of microbial transmission via coprophagy. This set of actinomycetes and their antimicrobials likely aid in gallery hygiene and consequently protect both an important nutrient source for *O. disjunctus* and their pupae. Furthermore, our findings demonstrate that the *O. disjunctus*/actinomycete system represents a tractable system for exploration of actinomycete specialized

metabolism at multiple scales, ranging from macroscale biogeography in natura to interactions of microbes at microscopic scales in vitro.

## Results

### Actinomycetes with antimicrobial properties are widespread in *O. disjunctus* galleries

Passalid beetles of the species *O. disjunctus* (formerly known as *Passalus cornutus*, and commonly referred to as 'bessbugs', *Figure 1A*) are widely distributed across eastern North America, where they are important decomposers of rotting timber (*Ceja-Navarro et al., 2014*; *Ceja-Navarro et al., 2019*; *Gray, 1946*; *Pearse et al., 1936*). This role has prompted interest in the *O. disjunctus* gut microbiota as a potential source of lignocellulose-processing microbes for biofuel efforts (*Ceja-Navarro et al., 2014*; *Ceja-Navarro et al., 2019*; *Nguyen et al., 2006*; *Suh et al., 2003*; *Suh et al., 2005*; *Urbina et al., 2013*). *O. disjunctus* is subsocial, with mating pairs establishing galleries within decaying logs where they rear their larvae (*Schuster and Schuster, 1985*; *Wicknick and Miskelly, 2009*). Large amounts of beetle frass accumulate within these galleries (*Figure 1B*). *O. disjunctus* is also coprophagic, and it is thought that microbes within frass continue digesting plant material as a kind of 'external rumen' between periods of consumption by the beetles (*Biedermann, 2020*; *Mason and Odum, 1969*; *Ulyshen, 2018*; *Valenzuela-González, 1992*). The frass is also notable as the adults feed it to the larvae, and parents and teneral siblings construct chambers from frass around metamorphosing pupa (*Biedermann, 2020*; *Gray, 1946*; *Schuster and Schuster, 1985*; *Valenzuela-González, 1992*; *Figure 1C*). Given the high nutrient content of frass and the complex parental behaviors associated with it, we drew parallels between this system and the other insect/actinomycete systems described above. Thus, we hypothesized that *O. disjunctus* galleries, and frass specifically, might contain actinomycete symbionts that have the potential to provide chemical defense to their host galleries and the food source on which their brood subsist.

To investigate if actinomycetes were associated with *O. disjunctus* galleries, we sampled material from 22 galleries across eastern North America (*Figure 1E*, *Supplementary file 1A*—Table S1). Samples included freshly produced frass from live beetles and larvae (as an indicator of their microbial gut content), and frass and wood from within the galleries. Pupal chamber material was also sampled when available, and in this case, pupae were also gently sampled with a swab. Using two selective media to enrich for actinomycetes, we isolated 339 bacterial strains (*Supplementary file 1B*—Table S2) and assayed their ability to inhibit growth of the Gram-positive bacterium *Bacillus subtilis* and the fungal pathogen *Candida albicans*. We found that the frequency of bioactivity was high among these isolates. Specifically, 76.1% of the collection displayed activity against *B. subtilis* and/or *C. albicans* (*Figure 1D*, *Supplementary file 1B*—Table S2), with 48.7% inhibiting both. The prevalence of actinomycetes displaying antimicrobial activity in vitro suggested that the *O. disjunctus*/actinomycete system might represent a rich environment for chemical ecology studies.

### In situ detection of microbial specialized metabolites

Next, we asked if specialized metabolites produced by actinomycetes could be detected directly in material from *O. disjunctus* galleries. To do so, we extracted frass and pupal chamber material with ethyl acetate, and analyzed the extracts using liquid chromatography coupled with high-resolution tandem mass spectrometry (LC-MS/MS). Surprisingly, we detected a wide array of microbial specialized metabolites in frass/pupal chamber material with identification levels of 1, 2, or 3 (see *Supplementary file 1C*—Table S3, and Materials and methods for identification criteria). Specifically, we detected 15 compounds that were grouped into seven distinct compound families based on high structural similarities (i.e., when analogs were grouped): the actinomycins D and $X_2$ (**1**, **2**); the angucyclinones STA-21 and rubiginone B2 (**3**, **4**); cycloheximide (**5**); the nactins monactin, dinactin, trinactin, and tetranactin (**7–10**); the polyene macrolides filipin III, filipin IV, and fungichromin (**13–15**); the polycyclic tetramate macrolactams (PTMs) alteramides A and B (**16**, **17**); and piericidin A (**24**) (*Figure 2A*; *Supplementary files 1C*—Table S3, 1D—Table S4; *Supplementary file 5*).

All of these families of compounds are known to be produced by actinomycetes and to have antimicrobial properties (*Gao et al., 2014*; *Hollstein, 1974*; *K'ominek, 1975*; *Mevers et al., 2017*; *Moree et al., 2014*; *Oka et al., 1990*; *Olano et al., 2014*; *Ortega et al., 2019*; *Protasov et al.,*

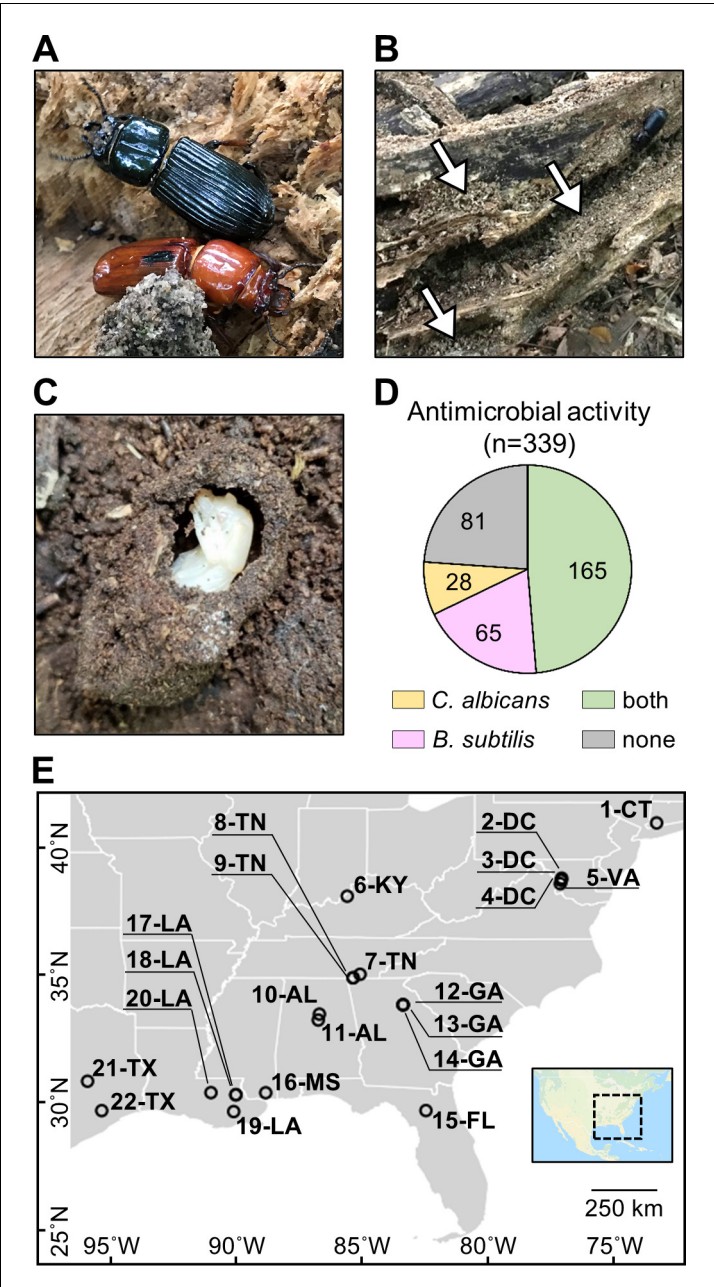

**Figure 1.** *O. disjunctus* beetles (**A**) inhabit and feed on decomposing logs. They build galleries that are filled with frass (**B**, white arrows are pointing to the frass), which is a central material in this system. This material is also used to build pupal chambers (**C**). We sampled 22 galleries across 11 states (**E**) and isolated actinomycetes from all samples. This actinomycete collection showed a high rate of antimicrobial activity against *B. subtilis* and/or *C. albicans* (**D**). Each gallery is represented on the map by a circle with the gallery code next to it. CT: Connecticut; DC: District of Columbia; VA: Virginia; KY: Kentucky; TN: Tennessee; GA: Georgia; FL: Florida; AL: Alabama; MS: Mississippi; LA: Louisiana; TX: Texas.

*2017*; *Shih et al., 2003*; *Song et al., 2005*; *Taniguchi et al., 2002*; *Urakawa et al., 1996*; *Zizka, 1998*). The average number of compound families detected per gallery was ~2.3, with only one gallery containing no detectable compounds, and four galleries containing four or five compound families (*Figure 2B*). Four families of compounds were detected in the pupal chamber material collected from gallery 17-LA: actinomycins, angucyclinones, polyene macrolides, and nactins. Interestingly, we also detected beauvericin, a compound with known insecticidal activity (*Wang and*

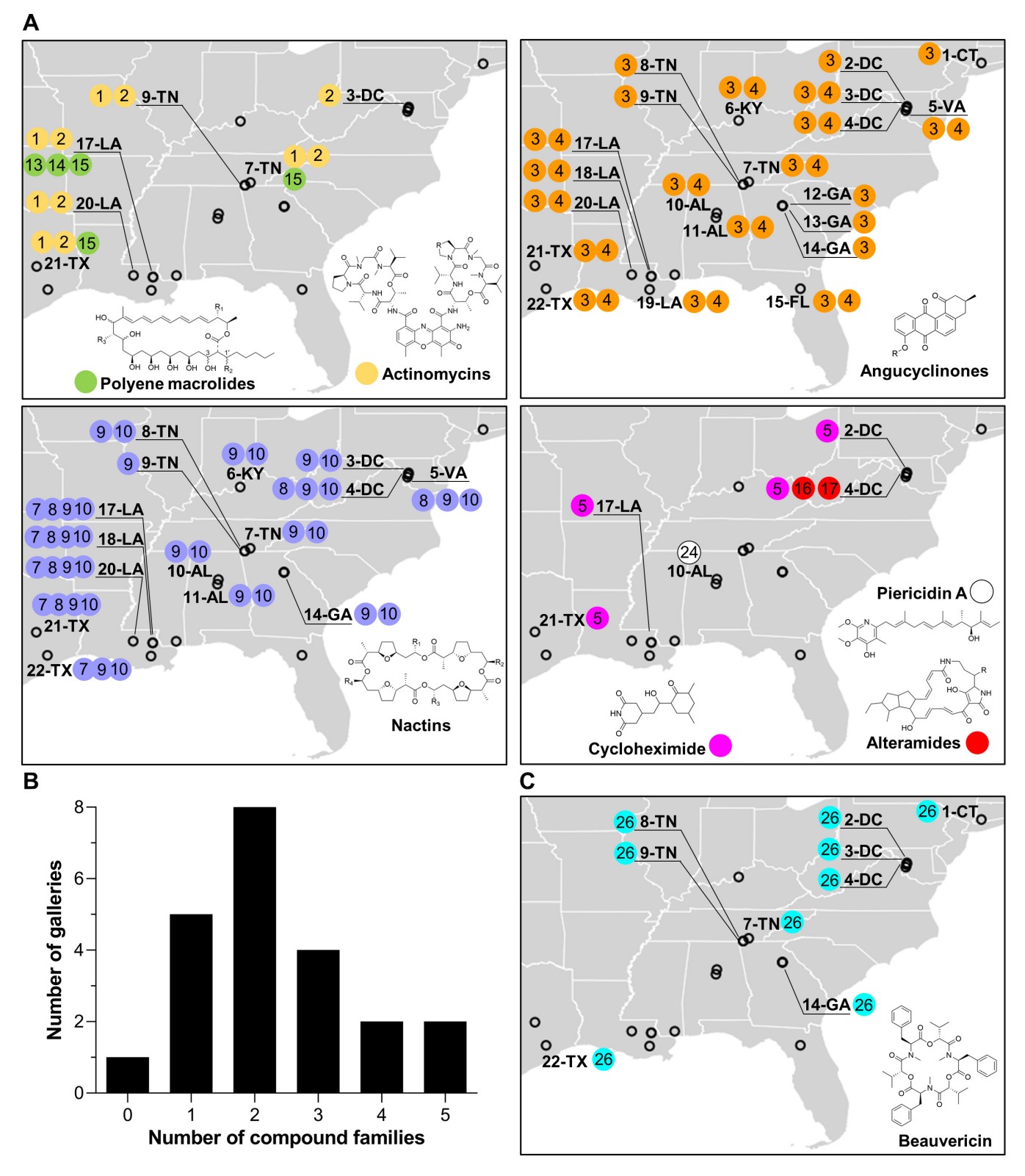

**Figure 2.** Geographic distribution of specialized metabolite families detected in frass material from wild beetle galleries. (A) Distribution of bacterially produced compounds. (B) Number of galleries in which 0–5 families of bacterially produced compounds were detected. (C) Distribution of beauvericin, a fungal metabolite. Numbers in circles represent the numeric code of each compound: actinomycin D (**1**), actinomycin X2 (**2**), STA-21 (**3**), rubiginone B2 (**4**), cycloheximide (**5**), monactin (**7**), dinactin (**8**), trinactin (**9**), tetranactin (**10**), alteramide A (**16**), alteramide B (**17**), piericidin A (**24**), and beauvericin (**26**).

*Figure 2 continued on next page*

*Figure 2 continued*

Each gallery is represented on the map by a circle with the gallery code next to it. CT: Connecticut; DC: District of Columbia; VA: Virginia; KY: Kentucky; TN: Tennessee; GA: Georgia; FL: Florida; AL: Alabama; MS: Mississippi; LA: Louisiana; TX: Texas.

*Xu, 2012*), in nearly half of the galleries (*Figure 2C*). Beauvericin is known to be produced by fungal entomopathogens like *Beauveria* spp. and *Fusarium* spp. (*Hamill et al., 1969*; *Logrieco et al., 1998*). Together, these results indicate that frass in *O. disjunctus* galleries commonly contains multiple types of antimicrobials produced by actinomycetes, and multiple antimicrobial molecules are found across the expansive geographic range of *O. disjunctus*. Beyond this, a molecule commonly produced by entomopathogenic fungi is also widespread in frass.

## Actinomycetes associated with *O. disjunctus* frass produce structurally diverse metabolites in vitro

We next sought to identify compounds produced by actinomycetes in our isolate library, with the dual goals of (i) identifying organisms that produce the metabolites seen in situ for further investigation and (ii) characterizing the chemical patterns across the isolates. To do so, we performed extractions from all the actinomycete cultures that produced zones of inhibition larger than 2 mm (a total of 161 strains) using ethyl acetate and submitted the crude extracts to LC-MS/MS analysis. Beyond the seven compound families detected in situ, we also identified isolates that produced antimycin A (**18**); the siderophore nocardamine (**19**), bafilomycins A1 and B1 (**20**, **21**), novobiocin (**22**), surugamide A (**23**), and nigericin (**25**) (see *Supplementary file 1E*—Table S5 and *Supplementary files 1–3* for details on the identification of individual compounds). With the exception of nocardamine, these compounds are also considered antimicrobials (*Kirby, 1956*; *Mahmoudi et al., 2006*; *Poulsen et al., 2011*; *Xu et al., 2017*). Other possible members of the actinomycins, angucyclinones, antimycins, PTMs, and surugamides were also detected, based only on similarities in the fragmentation pattern and exact mass (e.g., frontalamides, maltophilins, rubiginones). In total, we identified 25 compounds representing 12 distinct antimicrobial families plus 1 siderophore compound. We note that 14 of the compounds we identified here have been previously described to be produced by microbes associated with other insects (*Benndorf et al., 2018*; *Blodgett et al., 2010*; *Engl et al., 2018*; *Grubbs et al., 2020*; *Jiang et al., 2018*; *Kroiss et al., 2010*; *Mevers et al., 2017*; *Ortega et al., 2019*; *Poulsen et al., 2011*; *Schoenian et al., 2011*; *Seipke et al., 2011*; *Figure 3*). Collectively, these results reinforce the findings above that *O. disjunctus* frass plays host to actinomycetes that produce a rich array of antimicrobial compounds.

## Subsets of *O. disjunctus* frass isolates show patterns of stable association and recent/transient acquisition

In the course of characterizing the richness of compounds produced by frass isolates, we observed that actinomycetes from distant galleries often produced the same compounds in vitro (*Supplementary file 1B*—Table S2). Such a pattern could be explained either by intimate, sustained association of these actinomycetes with the beetle across the range of *O. disjunctus* or by frequent reacquisition of the actinomycetes that produce these specific compounds from the environments surrounding *O. disjunctus* galleries. To investigate this question, we built a phylogenetic tree of the actinomycete isolates from which we identified at least one compound. Since it is documented that the 16S rRNA gene, commonly used in bacterial phylogenetic studies, does not provide strong resolution for actinomycetes (*Choudoir et al., 2016*; *Guo et al., 2008*), we built a tree using concatenated sequences of 16S rRNA and those of three housekeeping genes (rpoB, gyrB, atpD) (see *Supplementary file 1F*—Table S6 for the GenBank accession number of each sequence). Duplicate strains were removed from the tree so as to not overrepresent clonal strains isolated from the same galleries, resulting in a total of 67 isolates placed on the tree. We defined duplicates as strains that were isolated from the same gallery that (i) have identical sequences for at least one of the four genes, (ii) have the same phenotype when growing on International *Streptomyces* Project 2 (ISP2-agar), and (iii) produce the same antimicrobial(s) in vitro.

Mapping the detected compounds onto a phylogenetic tree showed clear phylogenetic relationships associated with the production of specific compounds (*Figure 4*, *Figure 4—figure*

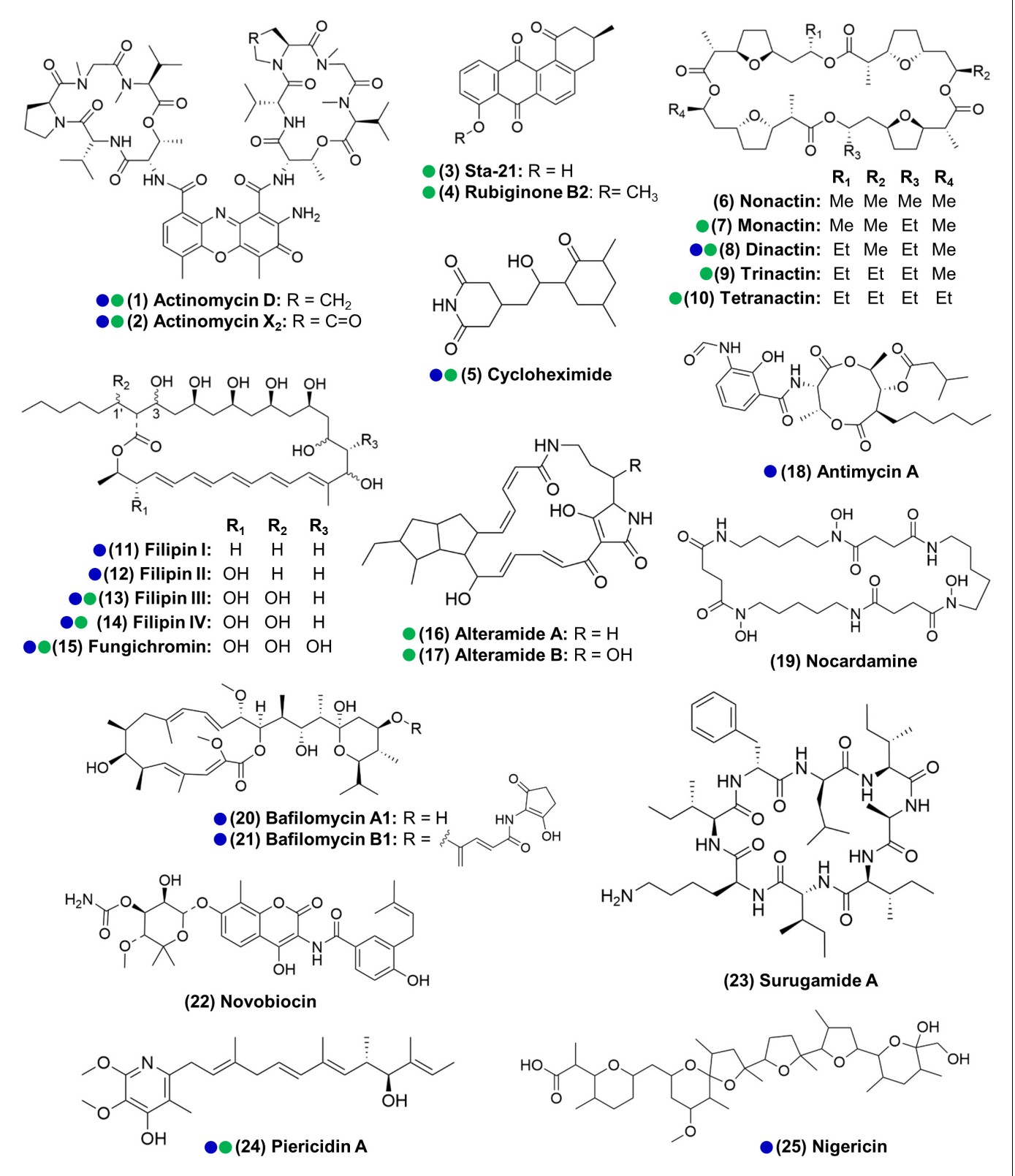

**Figure 3.** Actinomycetes associated with the *O. disjunctus* beetle produce structurally diverse specialized metabolites in vitro (1–25). Blue circles represent compounds that were previously described to be produced by microbes associated with other insects. Green circles represent compounds detected in the frass that was sampled from wild *O. disjunctus* galleries. Stereochemistry was assigned based on the commercial standard used or from the literature.

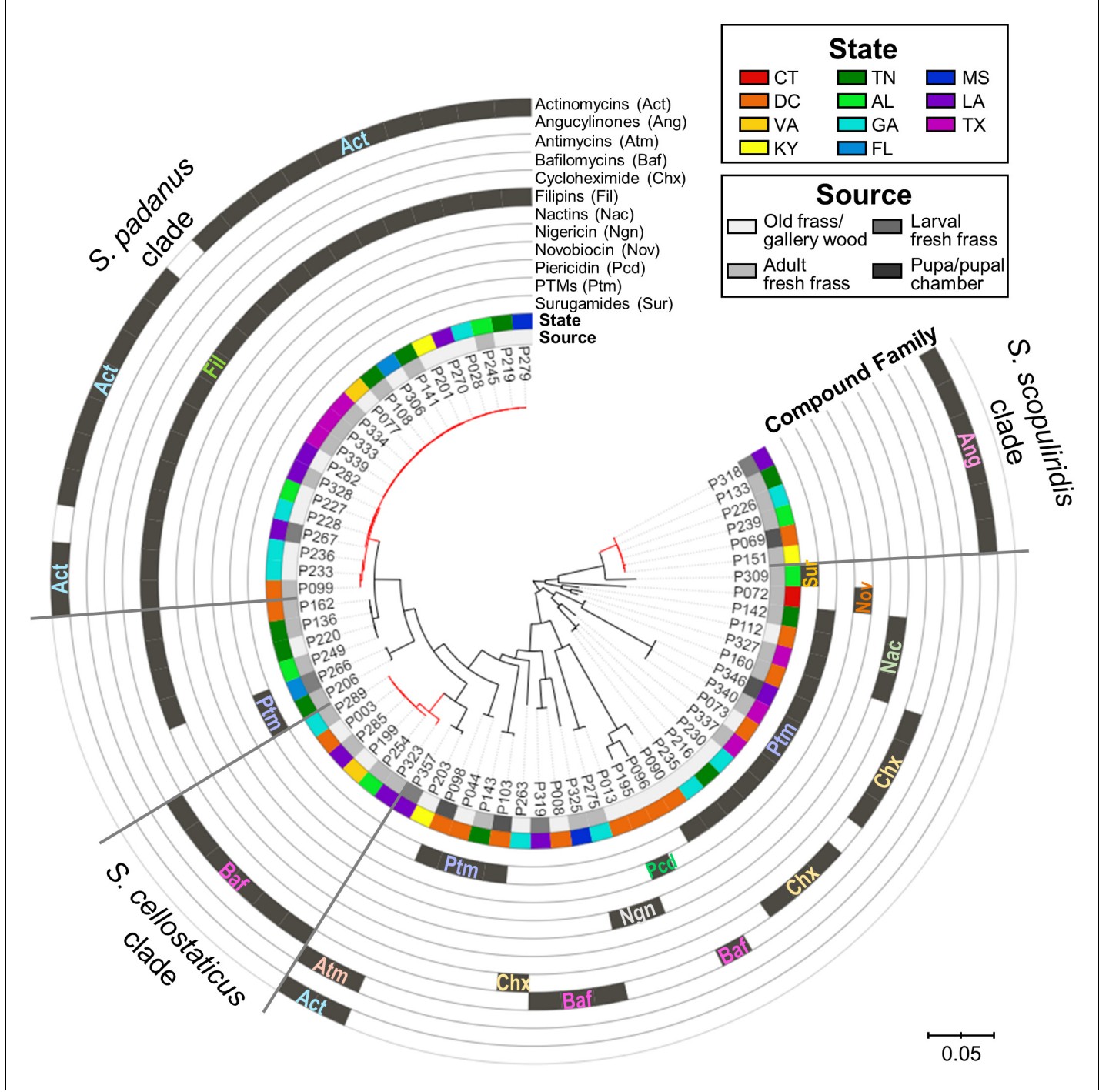

**Figure 4.** Maximum-likelihood phylogenetic tree built using concatenated sequences of four genes (16S rRNA, rpoB, gyrB, atpD) annotated with compounds produced by each microbial strain and their geographic and source origin (both represented by rings around the tree). Scale bar represents branch length in number of substitutions per site. The outgroup (*Mycobacterium tuberculosis* H37RV) was removed manually from the tree to facilitate visualization. See *Figure 4—figure supplement 1* for bootstrap values; *Figure 4—figure supplement 2* for a tanglegram comparing this tree with a chemical dissimilarity dendrogram; *Figure 4—figure supplements 3–5* for heatmaps showing the distance between strains geographic location; and *Figure 4—figure supplements 6–8* for a phylogenetic tree containing, respectively, duplicate strains that were removed in this main phylogenetic tree, *O. disjunctus* isolates plus *Streptomyces* isolated from soil samples, and *O. disjunctus* isolates plus *Streptomyces* isolated from tropical passalid beetles, termites, bees/wasps/ants, and soils. Branches in red highlight the three major clades: *S. padanus*, *S. cellostaticus,* and *S. scopuliridis*. Leaf labels represent the strain code. Act: actinomycins; Ang: angucyclinones; Atm: antimycins; Baf: bafilomycins; Chx: cycloheximide; Fil: filipins; Nac: nactins; Ngn: nigericin; Nov: novobiocin; Pcd: piericidin; Ptm: polycyclic tetramate macrolactams; Sur: surugamides.

*Figure 4 continued on next page*

*Figure 4 continued*

The online version of this article includes the following figure supplement(s) for figure 4:

**Figure supplement 1.** Maximum-likelihood phylogenetic tree built using concatenated sequences of four genes (16S rRNA, rpoB, gyrB, atpD).

**Figure supplement 2.** Tanglegram analysis comparing phylogenetic (left) and metabolic (right) information of *Streptomyces* strains associated with *O. disjunctus* shows chemo-evolutionary relationships among strains.

**Figure supplement 3.** Heatmap showing the distance in kilometers (in $\log_{10}$ scale) between the geographic origin of *Streptomyces* strains associated with *O. disjunctus* galleries.

**Figure supplement 4.** Heatmap showing the distance in degrees of latitude between the geographic origin of *Streptomyces* strains associated with *O. disjunctus* galleries.

**Figure supplement 5.** Heatmap showing the distance in degrees of longitude between the geographic origin of *Streptomyces* strains associated with *O. disjunctus* galleries.

**Figure supplement 6.** Maximum-likelihood phylogenetic tree built using the 16S rRNA gene sequence including duplicated strains, annotated with compounds produced by each microbial strain and their geographic and source origin (both represented by rings around the tree).

**Figure supplement 7.** Maximum-likelihood phylogenetic tree built using the 16S rRNA gene sequence of *Streptomyces* strains isolated from *O. disjunctus* and soil.

**Figure supplement 8.** Maximum-likelihood phylogenetic tree built using the 16S rRNA gene sequence of *Streptomyces* strains isolated from *O. disjunctus*, tropical passalid beetles, termites, bees/wasps/ants, and soil.

*supplement 6*). For example, actinomycins and filipins were consistently co-produced by a specific clade (bootstrap value of 100%) with high genetic relatedness, here identified as *Streptomyces padanus,* which was found in almost all galleries (19/22, *Supplementary file 6*). We also noted that distinct clades with high sequence similarity produced angucyclinones and bafilomycins, identified as *Streptomyces scopuliridis* and *Streptomyces cellostaticus*, respectively (bootstrap values of 100%). These clades are each composed of highly related strains despite being isolated from geographically distant galleries (as far as ~1900 km, 10 degrees of latitude, and 18 degrees of longitude apart; see *Figure 4—figure supplements 3–5*). Thus, we propose that these clades are likely stably associated with *O. disjunctus* throughout its range. Other isolates fell into areas of the tree that held higher phylogenetic diversity, and these strains produced a wider array of compounds, including cycloheximide, PTMs (e.g., alteramides), nigericin, piericidin, nactins, and novobiocin. The higher phylogenetic diversity of these isolates suggests that they represent transient members of the frass microbiota that have been more recently acquired from the environment, as opposed to being stably associated with *O. disjunctus*.

The relationship between the strain phylogeny and strain chemistry extended beyond the annotated compounds. A hierarchical clustering analysis was performed based on the chemical dissimilarity among culture extracts of the 67 strains, in which more than 19,000 chemical features were included. A tanglegram built between the phylogenetic tree and chemical dissimilarity dendrogram showed that many strains cluster together in both analyses (*Figure 4—figure supplement 2*). This result highlights the strong relationship between the phylogeny and metabolomes of these strains/ clades.

Notably, representatives of all three stable clades were isolated from the fresh frass, which serves as a proxy for the gut content, of adults and larvae. *S. scopuliridis* was also isolated from pupal chamber material, which is composed of frass (*Figure 4*, *Figure 4—figure supplement 6*). Thus, these species are associated with *O. disjunctus* across its life cycle. These findings support the notion that coprophagy could be a mode of transmission of these microbes since the larvae are thought to exclusively consume frass fed to them by adult beetles (*Valenzuela-González, 1992*). Additionally, an analysis of metagenomic data previously generated by members of our team confirmed that *Streptomyces* DNA is present along the adult *O. disjunctus* digestive tract and is enriched in the posterior hindgut (*Supplementary file 7*), the region in which remaining woody biomass is packed for its release in the form of frass. This finding further demonstrates that *Streptomyces* are normal members of the *O. disjunctus* gut microbiota.

To further investigate the diversity of the *O. disjunctus* streptomycete isolates, a phylogenetic analysis was performed using 16S rRNA gene sequences of 101 *Streptomyces* isolated from soil from diverse environments and locations worldwide, combined with the 16S rRNA sequences of the 67 *O. disjunctus* isolates investigated here (*Schlatter and Kinkel, 2014*). This tree shows that the *O. disjunctus* isolates encompass substantial diversity across the genus *Streptomyces*, but tended to group together in smaller clades with bootstrap values typically greater than 90 (*Figure 4—figure*

*supplement 7*). To further extend this analysis, we built another phylogenetic tree, this time including 16S rRNA gene sequences from more than 200 *Streptomyces* strains available in GenBank, which were isolated from the gut and galleries of other tropical passalid beetles from Costa Rica (*Vargas-Asensio et al., 2014*), the gut and exoskeleton of termites from South Africa (*Benndorf et al., 2018*), and different species of bees, ants, and wasps from Costa Rica (*Matarrita-Carranza et al., 2017*). This tree showed that many *O. disjunctus* isolates were grouped in clades with a high representation of strains isolated from a wide range of insects from distant locations (*Figure 4—figure supplement 8*).

Overall, these results are consistent with the idea that a subset of streptomycete clades that produce specific antimicrobials are stable inhabitants of *O. disjunctus* galleries and are likely transmitted across generations via coprophagy, while other clades are likely continually introduced from the surrounding woodland microbial communities. However, we note that deeper sampling could support a stable association for the more diverse clades as well. Additionally, the 16S rRNA gene sequences of many of the *Streptomyces* we isolated from *O. disjunctus* largely grouped with other insect-associated *Streptomyces* strains from around the world, indicating that they may be representatives of lineages that readily form stable relationships with insects.

## Specialized metabolites detected in situ show synergistic and antagonistic effects against a wild *Metarhizium anisopliae*

The fact that multiple microbial isolates from geographically remote galleries were found to produce the same compounds, together with the in situ detection of these compounds, indicates that antimicrobials made by actinomycetes often coexist in the frass environment. Therefore, we sought to explore chemical interactions (i.e., synergism and antagonism) between a subset of the most commonly identified molecules across our in situ and in vitro investigations. This list included the ionophore families of the nactins and filipins, the angucyclinone STA-21 (a Stat3 inhibitor; *Song et al., 2005*) and actinomycin X2 (a transcription inhibitor; *EL-Naggar et al., 1999*). During our fieldwork, we collected an *O. disjunctus* carcass that was partially covered with fungal biomass (Figure 6A). We identified this material as a strain of *M. anisopliae* (strain P287), an entomopathogenic fungus with a broad host range (*Zimmermann, 1993*).

We utilized *M. anisopliae* P287 as a target to investigate chemical interactions between the selected compounds. Using the Bliss Independence model (*Bliss, 1939*), we found multiple instances of compound interactions, including synergistic, antagonistic, and additive effects (*Figure 5*, *Figure 5—figure supplement 1*). Actinomycin X2 displayed robust synergism with both the filipins and the angucyclinone STA-21 (*Figure 5A, B*). The actinomycin X2/filipin result is notable since these compounds are usually made in concert by the same organism (*S. padanus*). In contrast, actinomycin X2 displayed an antagonistic effect when tested in combination with nactins (*Figure 5C*). The combination of filipins and STA-21 (*Figure 5D*) also showed a strongly antagonistic effect. Beyond this, the nactins displayed additive, synergistic, or antagonistic effects when combined with filipins or the angucyclinone STA-21, and these effects were concentration-dependent (*Figure 5—figure supplement 1*). Taken together, these results indicate that the rich chemical environment of frass is one in which synergism and antagonism among antimicrobials is likely commonplace.

## Actinomycetes growing directly in frass inhibit the growth of two strains of *M. anisopliae*, and *S. padanus* is a superior competitor

We next sought to develop an experimental system based on the *O. disjunctus* frass/streptomycete association that would enable the quantitative study of microbial interactions under environmentally relevant conditions. To do so, we selected three *Streptomyces* species that produced compounds that are abundant in *O. disjunctus* frass. These included *S. padanus* (P333, a producer of actinomycins and filipins), *S. scopuliridis* (P239, a producer of angucyclinones), and *Streptomyces californicus* (P327, a producer of nactins and the PTM alteramides). We also included two strains of *M. anisopliae* (*Supplementary file 8*): P287, used in the synergism/antagonism assays in the prior section, and P016, isolated from frass collected from a *O. disjunctus* gallery near Washington D.C.

We first tested the ability of the three streptomycetes to inhibit the two *M. anisopliae* strains in a plate-based assay. This assay showed that *S. padanus* P333 and *S. scopuliridis* P239 were able to produce robust zones of inhibition against both *M. anisopliae* strains, while *S. californicus* P327 did

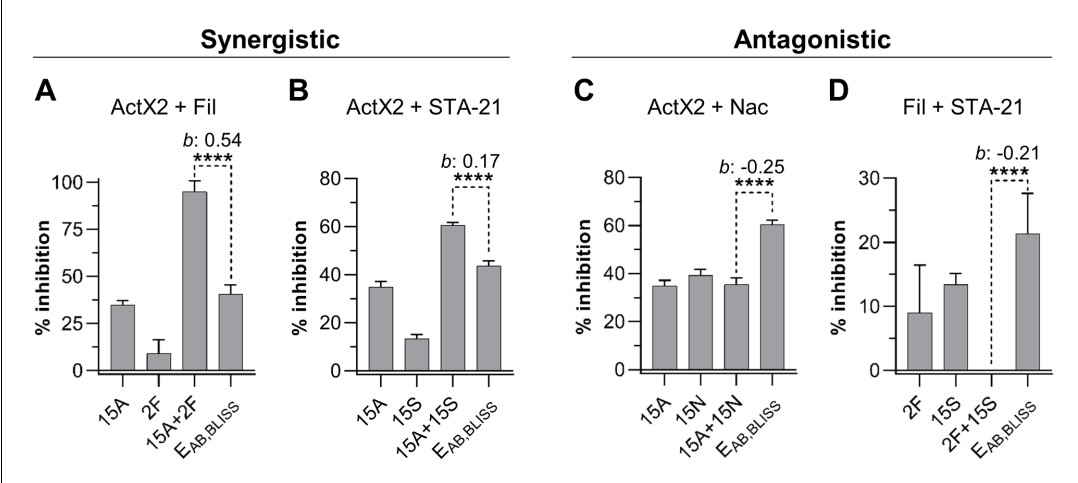

**Figure 5.** Actinomycin X2 (ActX2), filipins (Fil), nactins (Nac), and STA-21 display both synergistic (**A, B**) and antagonistic (**C, D**) interactions when tested for their ability to inhibit *M. anisopliae* P287 growth. Bars represent means (+ SD) of percent of growth inhibition (sample size: seven independent biological replicates). Statistical significance was measured using a t-test. ****p<0.0001. Numbers at the X-axis represent the tested concentration of each compound in µg/mL (F: filipins; A: actinomycin X2; N: nactins; S: STA-21). *b*: Bliss excess; $E_{AB,BLISS}$: expected value for an independent (additive) interaction between two drugs according to the Bliss Independence model. See *Figure 5—figure supplement 1* for other compound combinations. The online version of this article includes the following source data and figure supplement(s) for figure 5:

**Source data 1.** *M. anisopliae* P237 growth inhibition (in percent) in each one of the replicates of the compound interactions assay.
**Figure supplement 1.** Other compound combinations used in the compound interaction assay.

not (*Figure 6B*). We next asked whether or not these *Streptomyces* isolates could inhibit the growth of the *M. anisopliae* strains while growing in frass. To do so, we inoculated known quantities of spores of each microbe into microtubes containing 3 mg of sterilized dry frass. The water used as the inoculation vehicle supplied moisture, and the tubes were incubated at 30°C, which is close to the average temperature observed in *O. disjunctus* galleries (see *Supplementary file 1A*—Table S1), for 7 days. The microbes were inoculated in different combinations including (i) a single microbe per tube, (ii) one *Streptomyces* strain + one *M. anisopliae* strain, and (iii) combinations of two *Streptomyces* strains. Also, each microbe was inoculated into empty microtubes as a control to assess growth promoted by frass.

All microorganisms were able to use frass as a substrate for growth, including both *M. anisopliae* strains whose growth was enhanced ~14–20-fold compared to the no frass control (*Figure 6C*, *Figure 6—figure supplement 1*). We note that even though environmental frass often contains multiple antimicrobials (e.g., *Figure 2*), the heterogeneous nature of this material, plus autoclaving during preparation, likely means that any native antimicrobials were at low concentration and/or inactivated in these microbial growth assays. All three *Streptomyces* strains strongly inhibited *M. anisopliae* P016 and P287 growth in frass (p<0.001, *Figure 6D*). We next asked whether or not each *Streptomyces* strain produced its known antimicrobials while growing in these frass assays. Metabolomics analysis of crude extracts of the frass material revealed the presence of the actinomycins and filipins produced by *S. padanus*, nactins and alteramides produced by *S. californicus*, and angucyclinones produced by *S. scopuliridis*, matching the compounds produced in vitro by these three *Streptomyces* (*Figure 6F*, *Figure 6—figure supplements 2* and *3*). These results again highlight frass as an active site for production of antimicrobials, consistent with the notion that these molecules likely inhibit *M. anisopliae* growth. However, we note that other molecules not identified here could also play a role in this inhibition, as could competition for space and/or nutrients.

Next, we investigated if the *Streptomyces* strains were capable of inhibiting each other during growth on frass. When we co-inoculated pairs of *streptomyces* on frass, the growth of *S. padanus* P333 was not affected by either *S. californicus* P327 or *S. scopuliridis* (*Figure 6E*). However, *S. padanus* P333 strongly inhibited the growth of *S. californicus* P327. It was not possible to assay *S. scopuliridis* P239 growth via plate counts in the presence of the other *Streptomyces* due to its vulnerability to the antimicrobials they produced in vitro. However, we noted that production of the

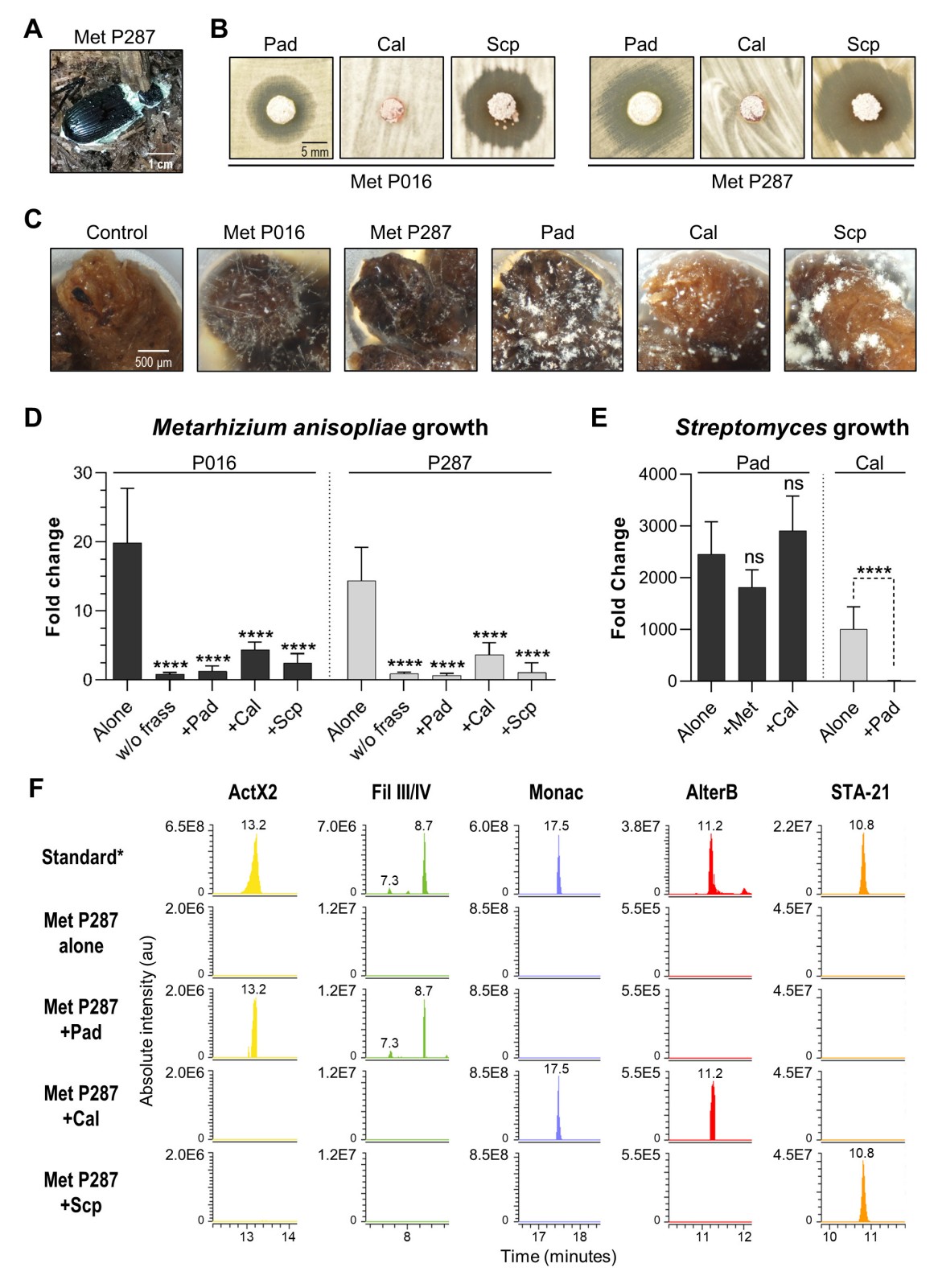

**Figure 6.** Competitive interactions between key *Streptomyces* isolates and entomopathogenic fungal strains directly in frass. (A) Beetle carcass with *M. anisopliae* P287. (B) Selected streptomycetes displayed a zone of inhibition against wild isolates of *M. anisopliae*. (C) Each selected microbe growing on frass material after 7 days of incubation (7× magnification). (D) *M. anisopliae* growth represented in fold change when growing alone versus in the presence of a streptomycete. (E) *S. padanus* P333 and *S. californicus* P327 growth represented in fold change when growing alone versus in the
*Figure 6 continued on next page*

*Figure 6 continued*

presence of another organism. (**F**) Extracted ion chromatograms of specialized metabolites detected in treatments containing *M. anisopliae* P287. *Standard: a mixture of crude ethyl acetate extracts of International *Streptomyces* Project 2 (ISP2)-agar cultures of the three streptomycetes. Pad: *S. padanus* P333; Cal: *S. californicus* P327; Scp: *S. scopuliridis* P239; Met: *M. anisopliae;* w/o frass: microbe added to an empty microtube; Alone: single microbe; ActX: actinomycin X2; FilIII/IV: filipins III and IV; Monac: monactin; AlterB: alteramide B. Bars represent means (+ SD) of growth in fold change from time 0 to day 7 of incubation (sample size: eight independent biological replicates). Statistical significance was measured by comparing treatments to microbes grown alone on frass using t-test when comparing two groups or ANOVA followed by Tukey's test when comparing more than two groups. ns: statistically not significant. ****p<0.0001. See *Figure 6—figure supplements 1–3* for additional results.

The online version of this article includes the following source data and figure supplement(s) for figure 6:

**Source data 1.** Growth in fold change of each microbe in each one of the treatments tested in the competition in frass assay.
**Figure supplement 1.** All microbes used in the interaction on frass assay were able to use the frass material as a substrate for growth.
**Figure supplement 2.** Extracted ion chromatograms of some specialized metabolites detected in treatments containing *M. anisopliae* P016.
**Figure supplement 3.** Extracted ion chromatograms of some specialized metabolites detected in treatments containing streptomycetes only.

angucyclinone STA-21, which is produced by *S. scopuliridis* P239, was dramatically reduced when *S. scopuliridis* P239 and *S. padanus* P333 were co-inoculated in frass, suggesting that *S. padanus* likely had a negative impact on *S. scopuliridis* P239 in this treatment (*Figure 6—figure supplement 3*). Collectively, these findings offer direct evidence that in frass *M. anisopliae* strains isolated from *O. disjunctus*-associated environments can be strongly inhibited by *Streptomyces* isolates that produce antimicrobials. Moreover, among the *Streptomyces* strains, *S. padanus* P333 appeared to be the superior competitor during co-cultivation on frass.

## Discussion

Symbioses in which insects partner with actinomycetes for chemical defense against pathogens are attractive models for investigating the ecology of specialized metabolites (*Behie et al., 2017*; *Chevrette et al., 2019*; *Chevrette and Currie, 2019*; *Matarrita-Carranza et al., 2017*; *Van Arnam et al., 2018*). The best-characterized examples of such symbioses include the eusocial, neotropical leafcutter ants and the solitary beewolf wasps, and their respective actinomycete associates (*Engl et al., 2018*; *Kaltenpoth et al., 2014*; *Li et al., 2018*; *Menegatti et al., 2021*). While these systems have provided a remarkable window into the ecology of specialized metabolism, many questions remain regarding insect/actinomycete symbioses in the context of differing social structures, mechanisms of microbial transmission, and strategies to combat pathogens.

Here, we investigated the subsocial passalid beetle *O. disjunctus*, and its frass, as a system for dissecting the chemical ecology of actinomycete specialized metabolism. Through direct chemical analysis, we found that frass from galleries across the geographic range of *O. disjunctus* contained at least seven different families of known actinomycete-produced antimicrobials, and that production of at least four of these families (actinomycins, angucyclinones, nactins, and cycloheximide) is widely distributed across eastern North America. Additionally, actinomycete isolates from frass produced 12 different families of antimicrobials, including all of those observed in frass. Using a simple assay with frass as a growth medium, we also demonstrated that multiple actinomycete isolates from *O. disjunctus* galleries could directly inhibit the growth of the fungal pathogen *M. anisopliae*. Taken together, these findings place the *O. disjunctus*/frass system among the most chemically rich insect/actinomycete associations characterized thus far. These results are consistent with a model in which this antimicrobial richness benefits *O. disjunctus* by limiting fungal pathogen growth in the frass inside its galleries. Beyond these findings, the tractability of the *O. disjunctus*/actinomycete system makes it an attractive model from multiple experimental perspectives, enabling research across scales from biogeographic surveillance to in vitro mechanistic investigation.

### The *O. disjunctus*/actinomycete partnership comprises a rich chemical system

A key obstacle in the study of the ecology of specialized metabolites is the detection of these compounds in situ. Historically, this detection has been challenging for multiple reasons: (i) specialized metabolites exist at relatively low concentrations within complex chemical environments (*Grenni et al., 2018*; *Kellner and Dettner, 1995*; *Schoenian et al., 2011*); (ii) their production

probably occurs in a dynamic spatio-temporal manner (*Debois et al., 2014*; *Pessotti et al., 2019*); (iii) additional factors like light, temperature, and pH might alter their chemical structures and stabilities (*Boreen et al., 2004*; *Cycoń et al., 2019*; *Edhlund et al., 2006*; *Gothwal and Shashidhar, 2015*; *Mitchell et al., 2014*; *Thiele-Bruhn and Peters, 2007*); and (iv) some compounds are likely degraded by surrounding microbes (*Barra Caracciolo et al., 2015*; *Gothwal and Shashidhar, 2015*; *Grenni et al., 2018*). For these reasons, knowledge of when and where microbes produce specialized metabolites in natural settings is extremely limited.

The frass found in *O. disjunctus* galleries constitutes an important commodity in the lifestyle of this beetle. The frass itself is composed of partially digested wood with high organic carbon content, and nitrogen fixation by microbes in the gut of *O. disjunctus* enhances its bioavailable nitrogen content as well (*Ceja-Navarro et al., 2014*; *Ceja-Navarro et al., 2019*). Thus, frass represents a valuable nutrient source for these beetles, any associated microbes, and potential pathogenic invaders. The frass is also a key component in this system because (i) all of the *O. disjunctus* individuals in a gallery are in constant contact with it, (ii) adults feed it to the larvae, and (iii) pupal chambers, which encapsulate fragile *O. disjunctus* pupae, are made of this material (*Pearse et al., 1936*; *Schuster and Schuster, 1985*). Based on parallels between *O. disjunctus* and other social insects that form actinomycete symbioses, we hypothesized that frass material was likely to contain actinomycete symbionts and their specialized metabolites.

To directly assess if antimicrobial compounds were present in this beetle/frass system, we sampled frass from 22 natural *O. disjunctus* galleries, and in some cases, we were also able to collect pupal chamber material (galleries 2-DC, 15-FL, and 17-LA). When we analyzed the chemical composition of these frass samples using high-resolution LC-MS/MS, we detected seven families of actinomycete-produced antimicrobials: actinomycins (**1**, **2**), angucyclinones (**3**, **4**), cycloheximide (**5**), nactins (**7–10**), polyene macrolides (**13–15**), piericidin A (**24**), and alteramides (PTMs) (**16**, **17**). Actinomycins, angucyclinones, nactins, and polyene macrolides were also detected in pupal chamber material (17-LA). Of these compounds, only actinomycin X2 and piericidin A have previously been directly detected in material associated with insects, for example, waste material of laboratory colonies of *Acromyrmex echinatior* (a species of Attine ant) and associated with beewolf antennal glands and cocoons, respectively (*Engl et al., 2018*; *Kaltenpoth et al., 2016*; *Kroiss et al., 2010*; *Ortega et al., 2019*; *Schoenian et al., 2011*).

This work expands the list of antimicrobials detected directly in material associated with insects to include the PTMs, polyene macrolides, cycloheximide, nactins, and angucyclinones. In addition, we found that *O. disjunctus* frass also commonly contains beauvericin, which is an insecticidal specialized metabolite known to be produced by multiple fungal entomopathogens. This observation, and our isolation of *M. anisopliae* from frass and an *O. disjunctus* carcass, suggests that *O. disjunctus* galleries are likely under pressure from fungal entomopathogens. Moreover, subsocial insects, such as *O. disjunctus*, are at higher risk than solitary insects of pathogenic spread due to their frequent social interactions (*Onchuru et al., 2018*). Based on these results, we hypothesized that the rich array of antimicrobials produced by actinomycetes in *O. disjunctus* frass affords these beetles defense against pathogenic overtake of both a food source and material used for protection during metamorphosis. It is important to note that *Streptomyces* spp. are also known producers of enzymes that degrade wood components, for example, cellulose, lignin, and xylose (*Book et al., 2014*; *Book et al., 2016*). Therefore, the streptomycete community associated with *O. disjunctus* may play a nutritional role in this system as well. Indeed, *Vargas-Asensio et al., 2014* provided strong evidence that *Streptomyces* play an important role as nutritional symbionts in tropical passalid beetles from Central America (*Vargas-Asensio et al., 2014*). However, we note that more research is necessary to further investigate this hypothesis specifically for *O. disjunctus*.

To lay the groundwork for hypothesis testing in this system, we developed an assay using sterilized frass as a growth medium. We used this assay to assess if *Streptomyces* spp. isolated from *O. disjunctus* galleries grew in this material, produced specialized metabolites, and inhibited the growth of entomopathogens. All three *Streptomyces* isolates we tested (including strains of *S. padanus*, *S. scopuliridis*, and *S. californicus*) grew, produced antimicrobials, and effectively curtailed pathogen (*M. anisopliae*) growth in frass. Additionally, *S. padanus*, which was the actinomycete most commonly isolated from frass, outcompeted the other two *Streptomyces* strains we tested, providing a rationale for its prevalence in *O. disjunctus* galleries. Together, our findings support a model in which diverse *Streptomyces* in *O. disjunctus* frass benefit this beetle by inhibiting the growth of

fungal pathogens in their galleries. Beyond this, these results illustrate that this simple, frass-based assay can be used to study interactions between microbes in this system in a nutrient environment similar (if not virtually identical) to that found in *O. disjunctus* galleries in nature. This assay sets the stage for further genetic/chemical experimentation to dissect the role of individual specialized metabolites that may regulate these microbial interactions.

## Contrasting chemical strategies across insect/actinomycete symbioses

The beewolf, leafcutter ants, and *O. disjunctus* systems may represent a spectrum of chemical defense strategies that are maintained by different modes of transmission and reflect distinct selective pressures. The beewolf system contains the highest level of symbiote specificity, with a single species (or species complex) of streptomycete symbiont, and a relatively low diversity of chemical scaffolds that vary in their relative concentrations (*Engl et al., 2018*; *Kaltenpoth et al., 2014*). Engl et al. hypothesized that this subtle variation in component concentrations within the beewolf antimicrobial cocktail has been sufficient to maintain its efficacy over evolutionary time due to the lack of a specialized antagonist (i.e., a specific pathogen that is encountered repeatedly over evolutionary time when beewolves construct their brood chambers) (*Engl et al., 2018*). In contrast, various species of leafcutter ants appear to have changed actinomycete partners multiple times throughout the history of their symbioses (*Cafaro et al., 2011*; *Li et al., 2018*; *McDonald et al., 2019*), which has likely led to increased diversity of associated antimicrobials found across and within species of leafcutter ants. Such a strategy makes sense given that leafcutter ants are in a constant arms race with a specific pathogen (*Escovopsis* sp.) that may evolve resistance over time (*Batey et al., 2020*).

The specialized metabolite richness we observe directly in frass from wild *O. disjunctus* galleries surpasses that described for beewolves and leafcutter ants. One possible explanation for this richness is that the unique vulnerabilities associated with the high nutritional content of frass, and the important role it plays in *O. disjunctus* social interactions, may place a premium on maximizing antimicrobial diversity in this material. This may be especially advantageous given that multiple types of opportunistic fungal pathogens, including *M. anisopliae* (based on isolations), and possibly *Beauveria* spp. and/or *Fusarium* spp. (based on detection of beauvericin), appear to be common residents in *O. disjunctus* galleries.

Unlike beewolves and leafcutter ants, *O. disjunctus* does not appear to have specialized structures for maintaining and transporting microbial symbionts (*Ulyshen, 2018*). Instead, we suggest that *O. disjunctus* relies on coprophagy for transmission of associated microbes across generations, which is a common mechanism for transfer of non-actinomycete symbionts in other insect systems (*Onchuru et al., 2018*). Our phylogenetic and chemical analysis of 67 *Streptomyces* isolates from *O. disjunctus* galleries indicates that at least three clades, including the *S. padanus*, *S. scopuliridis*, and *S. cellostaticus* clades, contain members that are highly related despite being isolated from across a wide geographic area. This pattern fits the expectation for symbionts that are likely transmitted to the progeny instead of randomly reacquired from the environment. The idea that coprophagy, which is the primary means of larval nutrient acquisition, may serve as a mechanism for microbial transmission is supported by our findings that multiple representatives of these clades were isolated directly from fresh frass produced by larvae and adult beetles. Beyond this, data from our previous metagenomic analysis indicates that *Streptomyces* inhabit the beetle digestive tract, with a notable enrichment in the posterior hindgut (*Supplementary file 7*). Our phylogenetic analysis also indicates that frass contains diverse actinomycetes that are transient or recently acquired members of this system.

Based on this evidence, the *O. disjunctus* system appears capable of maintaining both stable members and migrants that are constantly sampled from the surrounding environment. Thus, we suggest that in the case of *O. disjunctus* the relatively non-specific nature of coprophagy as a microbial transmission mechanism may enable multiple *Streptomyces* lineages to competitively inhabit this system, resulting in a correspondingly wide profile of antimicrobials with varied mechanisms of action. We note that further experiments will be required to directly verify if coprophagy is the main mode of transmission of associated actinomycetes in this system. Beyond this, a deeper chemical sampling of other insect/actinomycete systems will be required to determine if specialized metabolite richness similar to what we observe here for frass is typical for insects that employ coprophagy as a mode of vertical actinomycete transmission.

## Implications of synergy and antagonism in a system rich in antimicrobials

The high richness of antimicrobials found in *O. disjunctus* frass suggests that synergy or antagonism between these molecules may be commonplace in this environment. Strains of *S. padanus*, which we isolated from 19/22 of the galleries, typically produce both actinomycins and polyene macrolides (e. g., filipins), and these two antimicrobial families were also detected in frass from multiple galleries. When we tested actinomycin and filipin in combination against an *O. disjunctus*-associated strain of *M. anisopliae*, we found that they were strongly synergistic. Likewise, actinomycin X2 was also robustly synergistic with the most commonly detected antimicrobial in frass, STA-21 (an angucyclinone). Thus, synergism may potentiate the antimicrobial activity of multiple molecules produced by single strains, as well as molecules produced across species. These findings are aligned with previous work that has suggested that some insects, such as beewolves, might make use of cocktails of synergistic antimicrobials akin to 'combination therapy' (*Engl et al., 2018*; *Schoenian et al., 2011*). In contrast, we also found multiple instances of molecular antagonism, including between filipins and STA-21, and between actinomycin and nactins, which were the second most frequently detected antimicrobial in frass samples. While antagonism between molecules in frass may lead to diminished potency in the short term, emerging evidence indicates that antagonism can guard against the evolution of antimicrobial resistance (*Chait et al., 2007*). Collectively, our in vitro results, and the distributions of antimicrobials we detected in situ, lead us to speculate that the actinomycete community in frass likely produces an ever-shifting landscape of antimicrobial combinations, where their activities are constantly enhanced or dampened, but also buffered against the development of pathogen resistance. We hypothesize that such an environment may present a more challenging target for would-be pathogens compared to one in which a single antimicrobial, or antimicrobial combination, is dominant.

## *O. disjunctus*/actinomycete partnership as a model for investigating the biogeography of specialized metabolism

The detection of specialized metabolites directly in frass, combined with the expansive range of *O. disjunctus*, enabled us to study the biogeography of specialized metabolism within this system on a continental scale. Remarkably, four of the seven actinomycete compound families we detected in situ, including actinomycins, angucyclinones, nactins, and cycloheximide, were found throughout the range of *O. disjunctus,* with each compound being represented in colonies separated by >1900 km. Given the challenges associated with detecting specialized metabolites in situ, and our stringent thresholds for calling positive compound hits, we hypothesize that some compounds found at low frequency in our analyses are also likely to be widely distributed in *O. disjunctus* galleries. Taken together, these results indicate that the broad cocktail of antimicrobials collectively found in *O. disjunctus* galleries is consistently drawn from the same large molecular cohort over thousands of square kilometers, rather than being regionally limited. This wide geographic distribution of compounds is further supported by our in vitro analyses, with producers of actinomycins, polyene macrolides, bafilomycins, cycloheximide, and PTMs commonly isolated from colonies across the entire sampling area.

Notably, *Streptomyces* species that are candidates for stable members within this system (i.e., the *S. padanus*, *S. scopuliridis*, and *S. cellostaticus* clades) were found in galleries distributed across 10 degrees of latitude. Similarly, *S. philanthi* is tightly associated with beewolf wasps across an even greater latitudinal range (*Kaltenpoth et al., 2006*; *Kaltenpoth et al., 2014*). The wide latitudinal distribution of the *Streptomyces* species associated with these two insect hosts stands in contrast with patterns observed for soil-associated streptomycetes. Specifically, previous studies demonstrated that soil-dwelling streptomycetes, and specialized metabolite biosynthetic gene clusters in soil, were limited to much narrower latitudinal distributions (*Charlop-Powers et al., 2015*; *Choudoir et al., 2016*; *Lemetre et al., 2017*). Thus, results presented here, combined with the studies of beewolf wasps/*S. philanthi*, suggest that by partnering with insects, actinomycetes and their specialized metabolite arsenals can escape normally strong latitudinal constraints.

## Concluding remarks

The *O. disjunctus*/actinomycete system provides a new platform for investigation of the chemical ecology of specialized metabolites. Notably, this system enables investigation of patterns in microbial specialized metabolism associated with a single insect species in natura across scales ranging from thousands of kilometers to binary microbial interactions at micro scales in the laboratory. In contrast to archetypal insect/actinomycete symbioses that rely on highly specific symbiotes that produce a limited number of antimicrobial compounds, our results suggest that the *O. disjunctus* lifestyle enables both microbial and chemical richness in their galleries. Continued exploration of novel insect/actinomycete systems will be critical to gaining a complete understanding of the strategies and mechanisms that underpin evolutionarily durable chemical defenses against pathogenic microbes in natural settings.

# Materials and methods

## Chemical standards

Actinomycin D (Sigma-Aldrich, A1410), actinomycin X2 (Adipogen, BVT-0375), antimycin A (Sigma-Aldrich, A8674), bafilomycin A1 (Cayman Chemical, 11038), bafilomycin B1 (Cayman Chemical, 14005), cycloheximide (ACROS Organics, AC357420010), filipin complex (Sigma-Aldrich, F9765), nigericin sodium salt (Cayman Chemical, 11437), nactins mixture (Cayman Chemical, 19468), novobiocin sodium salt (Calbiochem, 491207), piericidin A (Cayman Chemical, 15379), rubiginone B2 (Santa Cruz Biotechnology, sc-212793), and STA-21 (Sigma-Aldrich, SML2161).

## Culture media

International *Streptomyces* Project 2 (ISP2)-broth: malt extract 10 g/L, yeast extract 4 g/L, dextrose 4 g/L, pH 7.2. ISP2-agar: ISP2-broth plus agar 18 g/L. Adapted AGS: L-arginine 1 g/L, glycerol 12.5 g/L, NaCl 1 g/L, $K_2HPO_4$ 1 g/L, $MgSO_4.7H_2O$ 0.5 g/L, $FeSO_4.7H_2O$ 10 mg/L, $MnCl_2.4H_2O$ 1 mg/L, $ZnSO_4.7H_2O$ 1 mg/L, and agar 18 g/L, pH 8.5. Potato dextrose broth (PDB): potato starch 4 g/L and dextrose 20 g/L. Potato dextrose agar (PDA): PDB plus agar 18 g/L. $0.1\times$ PDB+MOPS: potato starch 0.4 g/L, dextrose 2 g/L, 0.165 M MOPS, pH 7.0. SM3: dextrose 10 g/L, peptone 5 g/L, tryptone 3 g/L, sodium chloride 5 g/L, and agar 18 g/L, pH 7.2. V8-juice-agar: V8-juice 20% (v/v), agar 18 g/L. All media were autoclaved for 30 min at 121°C and 15 psi.

## Environmental sample collection

Fallen decaying logs were pried open to initially examine if *O. disjunctus* beetles were present. Ethanol-cleaned spatulas were used to obtain three samples of frass ('old frass') and wood from different parts of the galleries. Whenever found, pupal chambers were scraped out of the galleries with a clean spatula, the pupa was placed back inside the gallery, the pupal chamber material was placed in a clean plastic bag, and the pupa was gently swabbed with a sterile cotton swab. Beetles (3–6) were temporarily collected and placed inside a sterile Petri dish for 15–30 min to allow them to produce frass ('adult fresh frass'), and this material was transferred to a sterile microtube utilizing a clean set of tweezers; the same was performed with larvae when they were found in the galleries ('larval fresh frass'). Upon completion of sampling, logs were placed back into their original position to minimize environmental impact. Samples were shipped the same day to our laboratory, according to the United States Department of Agriculture (USDA) guidelines and stored at 4°C for up to 1 month before being processed. A small part of each sample (3–5 mg) was stocked in 25% glycerol at −80°C for microbial isolation, and the rest was set aside for chemical extraction. In total, 22 galleries were sampled across 11 US states. Check *Supplementary file 1A*—Table S1 for geographic location and abiotic information of each gallery. The following permits were acquired prior to sample collection and transportation: USDA Permit to Move Live Plant Pests, Noxious Weeds, and Soil: P526P-18-03736; Rock Creek National Park permit number: ROCR-2018-SCI-0021; Jean Lafitte National Park permit number: JELA-2019-SCI-0001. For all other locations, the access was granted on the same day of collection by either the owner or manager of the property. The state boundaries in the map containing the geographic location of each sampled gallery were plotted using the 2019 TIGER/Line Shapefiles provided by the U.S. Census Bureau 2019 (accessed on March/2020, https://catalog.data.gov/dataset/tiger-line-shapefile-2019-nation-u-s-current-state-and-equivalent-national).

## Isolation of microorganisms

An aliquot of frass, wood, and pupal chamber samples stored in 25% glycerol was spread with glass beads onto two selective media: SM3 and AGS supplemented with cycloheximide (10 µg/mL) and nalidixic acid (50 µg/mL) to enrich for actinomycetes. Swabs containing pupal samples were swabbed onto the same media. The beetle carcass found in the environment (*Figure 6A*) was dissected, and the fuzzy material collected was spread onto PDA plates for isolation of fungi. Plates were incubated at 30°C and periodically checked for microbial growth for up to a month. Microbial colonies were picked, streak-purified, and stocked in 25% glycerol at −80°C. Check *Supplementary file 1B*—Table S2 for detailed information about each isolated strain.

## Antagonism assays

All microbial isolates were grown on ISP2-agar medium for 1 week at 30°C for antimicrobial assays and chemical extraction. After 7 days of incubation, a plug-assay was performed: 5 mm plugs were transferred from the culture plates to the ISP2-agar plates containing a fresh lawn of *B. subtilis* or *C. albicans* (1 day before the assay, the indicator strains *B. subtilis* 3610 and *C. albicans* GDM 2346 WT were grown in 5 mL of ISP2-broth for 16–18 hr at 30°C, 200 rpm; both indicator strains were then diluted 1:50 in fresh ISP2-broth and spread onto a new ISP2 plate with a swab to create a lawn). Plates were incubated at 30°C for 24 hr. Activity was visually inspected by measuring the zone of inhibition (ZOI). In some specific cases, *M. anisopliae* strains P016 and P287 were used as the indicator strain. A lawn of these fungi was created by spreading spores with a swab onto ISP2-agar plates, and in this case the incubation time was 3 days before measuring the ZOI. Spores were collected from 7-day-old *M. anisopliae* P016 and P287 growing on V8-juice-agar plates at 25°C under constant light.

## Chemical extractions

### Cultures on ISP2-agar

Ten 5 mm plugs were collected from culture plates and placed in a 2 mL microtube with 750 µL of ethyl acetate, sonicated for 10 min, and left at room temperature (RT) for 1 hr. The solvent was then transferred to a new microtube and dried under vacuum at 45°C. An extraction control of sterile ISP2-agar plates was performed following the same steps.

### Environmental samples

Frass and pupal chamber material samples were extracted three times in ethyl acetate: 10 mL of ethyl acetate was added to 4–5 g of material placed in a 50 mL conical tube, sonicated for 10 min, placed on a rocking shaker at 60 rpm for 30 min, and decanted. The obtained extract was centrifuged at 5000 rpm for 5 min to pellet the remaining frass material, and the solvent was dried under vacuum at 45°C. An extraction control without any sample added to the tube was performed following the same steps.

## LC-MS and LC-MS/MS analysis

Crude extracts were resuspended at 1 mg/mL in 500 µL of methanol containing an internal standard (reserpine at 1 µg/mL), sonicated for 5 min, and centrifuged for another 5 min at 13,000 rpm to pellet particles. A 50 µL aliquot was taken from each sample and pooled to generate the pooled-QC for quality control. Extracts were analyzed in a randomized order using an ultra-high-pressure liquid chromatography system (LC, Thermo Dionex UltiMate 3000, Thermo Fisher, USA) coupled to a high-resolution tandem mass spectrometer (MS/MS, Thermo Q-Exactive Quadrupole-Orbitrap, Thermo-Fisher) equipped with a heated electrospray ionization source (LC-MS/MS), using a C18 column (50 mm × 2.1 mm, 2.2 µm, Thermo Scientific Acclaim RSLC). A gradient of 0.1% formic acid in water (A) and 0.1% formic acid in acetonitrile (B) was used at a flow of 0.4 mL/min, specifically: 0–1 min 30% B, 1–13 min 30–100% B, 13–16.5 min 100% B, 16.5–17 min 100–30%, and 17–20 min 30% B. The injection volume was 5 µL, and the column oven was set at 35°C. Analyses were performed in profile mode both with and without MS/MS acquisition (LC-MS/MS and LC-MS, respectively). The full MS1 scan was performed in positive mode, resolution of 35,000 full width at half-maximum (FWHM), automatic gain control (AGC) target of $1 \times 10^6$ ions, and a maximum ion injection time (IT) of 100 ms, at a mass range of *m/z* 200–2000. For LC-MS/MS analysis, the MS/MS data was acquired

using the data-dependent analysis mode (DDA), in which the five most intense ions were sent for fragmentation (Top5 method), excluding repetitive ions for 5 s (dynamic exclusion), at a resolution of 17,500 FWHM, AGC target of $1 \times 10^5$ ions, and maximum IT of 50 ms, using an isolation window of 3 $m/z$ and normalized collision energy of 20, 30, and 40. LC-MS runs of environmental samples were performed in three technical replicates aiming to increase the confidence in the observed chemical features. In some cases, a targeted LC-MS/MS method was optimized to confirm the presence of the annotated compound. The raw data was deposited on the Mass Spectrometry Interactive Virtual Environment (MassIVE, https://massive.ucsd.edu/, identifiers: MSV000086314, MSV000086312, MSV000086311, MSV000086330, MSV000086423).

## Compound identification

The LC-MS/MS data collected was processed using the open-access software MS-Dial version 4.0 (*Tsugawa et al., 2015*) using optimized parameters. Chemical dereplication was performed by comparing the $m/z$ of detected features to the databases Antibase 2012 (*Laatsch, 2012*) and the Dictionary of Natural Products (http://dnp.chemnetbase.com/, accessed on February 2019), allowing a maximum mass accuracy error of ±5 ppm, and checking for the presence of at least two adducts with similar retention time (±0.1 min). Molecular networking using the GNPS platform (*Wang et al., 2016*) was also performed for de-replication. The extracted ion chromatogram (EIC) of each chemical feature with a hit in one of the databases was inspected manually in order to evaluate the peak quality using a $m/z$ range allowing a mass error of ±5 ppm. For environmental samples, chemical features of interest were validated by checking for their presence in both LC-MS/MS run and three LC-MS runs. Each database hit was further confirmed at different levels according to *Sumner et al., 2007*. For both culture and environmental extracts, a level 1 identification was assigned when both the retention time and fragmentation pattern were matched with a commercial standard. In the absence of a commercial standard, a level 2 identification was assigned by matching the MS2 spectrum with spectra available in the literature or in the GNPS spectral library. A level 3 identification was assigned based on spectral similarities of the compound and a commercially available standard of an analog compound. In cases in which a MS2 was not detected in a given environmental sample, a hit was considered real only if the retention time and mass accuracy were within our tolerance levels (±0.1 min, ±5 ppm error), and if either two adducts were detected and/or other members of the same family of compounds were also detected in the same sample, we consider such IDs level 2 or 3 depending on the availability of standard spectra. If a MS2 was not detected at all in any of the environmental samples, it was not considered a real hit even if it passed all the criteria above. Compounds of the polyene macrolides family, which are known for being unstable (**5**), were challenging to annotate due to their low peak height. For this reason, our criteria for polyene macrolides annotation in environmental samples were retention time and mass accuracy within our tolerance levels (±0.1 min, ±5 ppm error), presence of at least two adducts in one of the technical replicates, and presence of at least one adduct in two other technical replicates.

## Multi-locus sequence analysis and phylogenetic tree construction

Four genes of selected microbial strains were partially sequenced (16S rRNA, gyrB, rpoB, atpD). Strains were grown in 5 mL of ISP2-broth at 30°C, 200 rpm for 1–7 days. Cultures were centrifuged at 5000 rpm for 5 min, supernatant was removed, and the pellet was washed with double-distilled water. Pellet was then extracted using the DNeasy Blood and Tissue kit (Qiagen, 69504) following the manufacturer's instructions for Gram-positive bacterial samples. The 16S rRNA gene was amplified using the 27F (5′-AGAGTTTGATCCTGGCTCAG-3′) and 1492R (5′-GGTTACCTTGTTACGACTT-3′) primers using parameters as follows: 98°C for 30 s followed by 34 cycles of 98°C for 5 s, 58°C for 30 s, 72°C for 45 s, and finalized with 72°C for 5 min. Other genes were also amplified and sequenced: rpoB (primers: 5′-GAGCGCATGACCACCCAGGACGTCGAGGC-3′ and 5′-CCTCGTAG TTGTGACCCTCCCACGGCATGA-3), atpD (primers: 5′-GTCGGCGACTTCACCAAG GGCAAGGTG TTCAACACC-3′ and 5′-GTGAACTGCTTGGCGACGTGGGTGTTCTGGGACAGGAA-3), and gyrB (primers: 5′-GAGGTCGTGCTGAC CGTGCTGCACGCGGGCGGCAAGTTCGGC-3′ and 5′-GTTGATG TGCTGGCCGTCGACGTCGGCGTCCGCCAT-3) following the same procedures, except that the annealing step was held at 70°C for 30 s. All PCR reactions were performed using the Phusion Green Hot Start II High-Fidelity PCR Master Mix (Fisher Scientific, F566S). For reactions where gel

extraction was necessary, the QIAquick Gel Extraction Kit (Qiagen, 28704) was used. The PCR or gel extracted products were sent for DNA cleanup and sequencing at the UCB DNA Sequencing Facility (https://ucberkeleydnasequencing.com/) using the same primers, with the exception of gyrB (sequencing primers: 5′-GAGGTCGTGCTGACCGTGCTGCA-3′ and 5′-CGCTCCTTGTCCTCGGCC TC-3). The resulting forward and reverse sequences were assembled and trimmed on Geneious R9 (*Kearse et al., 2012*) (allowing an error probability limit of 1%) to generate a consensus sequence. A BLAST search on GenBank (*Sayers et al., 2020*) was performed to find the closest related species. All trimmed sequences are available on GenBank (see *Supplementary file 1F*—Table S6 for accession numbers). Consensus sequences of each gene were aligned separately with *Mycobacterium tuberculosis* H37RV respective gene (used as the outgroup) on Geneious 9.1.8 (*Kearse et al., 2012*) using MUSCLE (*Edgar, 2004*) default parameters and trimmed at the same position in both ends. The four trimmed sequences obtained for the same strain were concatenated (gyrB-rpoB-16S-atpD). The final concatenated sequences of the 67 selected microbes plus *M. tuberculosis* H37RV were aligned again using the same parameters. This alignment was used to build a maximum-likelihood phylogenetic tree using IQTree (*Minh et al., 2020*) with the best-fit model chosen as GTR+F+R4, and using 1000 bootstrap repeats to estimate the robustness of the nodes. The tree was visualized, customized, and annotated using the Interactive Tree of Life program (iTol) (*Letunic and Bork, 2019*) v.5.6.3 (https://itol.embl.de). Trees including 16S rRNA gene sequences from other studies were built in the same way using the *Streptomyces* sequences deposited on GenBank (PopSet numbers: 1095870380, 702102129, 663498531, 1139695609).

## Heatmap, chemical dissimilarity dendrogram, and tanglegram analysis

The heatmaps were plotted in Python using the Matplotlib library v.3.1.1. The chemical dissimilarity dendrogram was built using hierarchical cluster analysis (HCA) in Rstudio software v.1.4.1103. First, the chemical features table was filtered to remove features detected in the blank samples (methanol and medium extract), then the remaining features were converted into a presence/absence format, using peak intensity higher than $1 \times 10^6$ as the threshold. A Jaccard distance matrix was generated using the Vegan package, 'vegdist' function (v.2.5-7, https://cran.r-project.org/web/packages/vegan/). This matrix was used to build a dendrogram through a HCA, using the Cluster package (v.2.1.1, https://cran.r-project.org/web/packages/cluster/), 'hclust' function with the 'average' agglomeration method (=Unweighted Pair Group Method with Arithmetic Mean). The chemical dissimilarity dendrogram was then compared in a tanglegram to the phylogenetic tree using the Dendextend package, 'tanglegram' function (v.1.14.0, cran.r-project.org/web/packages/dendextend/). The generated tanglegram was untangled using the untangle function with the 'step2side' method, and the entanglement value was obtained using the 'entanglement' function.

## Compound interaction assay

Interaction between compounds was assessed using *M. anisopliae* P287 as an indicator. *M. anisopliae* P287 was grown on V8-juice-agar at 25°C under constant light for 7 days, spores were collected with a loop, resuspended in 0.03% Tween80, and filtered through a cheesecloth. The concentration of spores in the inoculum was estimated using a hemocytometer and adjusted to $3–5 \times 10^5$ spores/mL. In order to validate the spores count, dilutions of the spores solution were plated on PDA plates and incubated at 25°C under constant light for 3 days to count colony forming units (CFUs). The assay was performed in a final volume of 100 μL/well of 0.1× PDB+MOPS medium in 96-well plates, containing $3–5 \times 10^4$ spores/mL. Selected compounds were tested alone and in pairwise combinations in seven biological replicates. The concentrations tested were as follows: actinomycin X2 and nactins: 15 μg/mL; STA-21: 15 μg/mL and 20 μg/mL; filipins: 2 μg/mL and 4 μg/mL. Antibiotic stocks solutions were prepared in DMSO and diluted in 0.1× PDB+MOPS to a final concentration of 0.6% or 0.7% DMSO in the well. Solvent, solo inocula, and medium sterility controls were added separately into seven wells in a 96-well plate, each one becoming an independent replicate. The solvent control was composed of spores, medium, and 0.6 or 0.7% DMSO; the inoculum control (IC) was composed of spores in medium; and the medium sterility control (MC) was composed of medium only. Plates were incubated at 30°C for 48 hr. At this time 0.002% (w/v) of the redox indicator resazurin was added (prepared in double-distilled $H_2O$ and filter-sterilized), and plates were incubated

again for another 24 hr. Fluorescence of the redox indicator was measured at 570 nm and 615 nm for excitation and emission, respectively, using a plate reader (SpectraMax i3x, Molecular Devices).

The type of compound interaction was determined by calculating the Bliss predicted value for independent effect ($E_{AB,Bliss}$) and Bliss excess ($b$) followed by a t-test using a method described elsewhere (*Folkesson et al., 2020*) with some modifications. The effect of each treatment was evaluated calculating the fractional inhibition (FI) when compared to the inoculum control.

$$FI = 1 - (T_F - MC_F)/(IC_F - MC_F)$$

$T_F$: fluorescence intensity of the treatment
$MC_F$: fluorescence intensity of the medium control
$IC_F$: fluorescence intensity of the inoculum control

The percentage of inhibition was calculated by multiplying the FI by 100. The type of compound interactions was divided into three categories: synergistic, antagonistic, and additive, and it was determined using the Bliss Independence model following the methods described elsewhere (*Folkesson et al., 2020*) with some modifications. Since we chose to report our data based on the observed growth inhibition and not survival, the formula was adjusted according to the Bliss Independence model (*Bliss, 1939*). Therefore, the Bliss expected value for an independent (additive) effect ($E_{AB,BLISS}$) and Bliss excess ($b$) were calculated as follows:

$$E_{AB,Bliss} - FI_A + FI_B - FI_A FI_B b = E_{AB,BLiss} - FI_{AB}$$

where A and B represent the compounds tested ($FI_A$ and $FI_B$: compounds tested alone, $FI_{AB}$: compounds tested in combination). Therefore, $E_{AB,BLISS}$ represents the expected FI value if the compounds have an additive effect (according to Bliss Independence model), whereas $FI_{AB}$ represents the actual value observed when compounds were tested in combination.

The $E_{AB,Bliss}$ was calculated using the replicates of each compound in all possible combinations, generating several FI expected values for each compound pairwise combination. On the other hand, the Bliss excess was calculated using average numbers. A t-test was performed to compare the means of $E_{AB,Bliss}$ and $FI_{AB}$. Pairwise combinations with $b \geq 0.08$, $0.08 \leq b \geq -0.08$, and $b \leq -0.08$ were classified as synergistic, additive, and antagonistic, respectively, when the p-value was $\leq 0.05$.

## Interaction on frass assay

Pieces of frass (3 mg) were placed inside 200 µL microtubes, autoclaved, and oven-dried. Spores of selected microbes were inoculated in 15 µL ($0.4$–$2.8 \times 10^3$ CFU of each microbe per microtube) in different combinations. Each combination was tested separately in eight different microtubes (biological replicates): (i) a single microbe per tube, (ii) one streptomycete + *M. anisopliae*, and (iii) two streptomycetes. Each microbe was also added to empty microtubes as a growth control, and some microtubes containing frass were inoculated with sterile water as a sterility control. In the case of multiple microbes per tube, spores of each microbe were pre-mixed before adding them to the frass. Therefore, each treatment had its own initial inoculum, which was plated to verify the exact initial concentration of each microbe in each treatment by CFU count. All tubes were vortexed for 3 s and spun down for another 3 s. Microtubes were then incubated at 30°C for 1 week. After the incubation time, 100 µL of a solution of 0.03% of Tween80 was added to each tube and vortexed for 30 s, left at RT for 1 hr, and vortexed again for another 30 s to detach cells from the frass. An aliquot of each tube was serially diluted and plated for CFU count. The rest of the material was extracted with ethyl acetate (aqueous phase) and methanol (frass material). Crude extracts were submitted for metabolomics analysis using the same pipeline described above. In both cases of initial and final CFU counts, *M. anisopliae* counts were performed using PDA plates supplemented with apramycin (25 µg/mL) to suppress the growth of the co-inoculated streptomycete, and streptomycetes counts were performed using ISP2-agar plates.

## Acknowledgements

We thank the staff at Rock Creek and Jean Lafitte national parks for granting us the permission to collect samples inside the parks; Michael E Rush, Terry Presswood, Theresa Presswood, and Janice

Tharaldson for granting us access to their private properties and for the support given during sample collection; Daniel and Robbie Williams, and their family, especially Caden Williams, for helping us locate and collect preliminary samples on their land; and Cindy Pino, Vineetha Zacharia, and Carlos E R Machado for their support during the fieldwork. Part of this work was performed at the Lawrence Berkeley National Laboratory under the US Department of Energy contract number DE-AC02-05CH11231.

This manuscript has been authored by an author at Lawrence Berkeley National Laboratory under Contract No. DE-AC02-05CH11231 with the U.S. Department of Energy. The U.S. Government retains, and the publisher, by accepting the article for publication, acknowledges, that the U.S. Government retains a non-exclusive, paid-up, irrevocable, world-wide license to publish or reproduce the published form of this manuscript, or allow others to do so, for U.S. Government purposes.

## Additional information

### Funding

| Funder | Grant reference number | Author |
|---|---|---|
| University of California Berkeley | startup funds | Rita de Cassia Pessotti<br>Bridget L Hansen<br>Jewel N Reaso<br>Laila El-Hifnawi<br>Matthew F Traxler |
| Searle Scholars Program | SSP-2016-1411 | Rita de Cassia Pessotti<br>Bridget L Hansen<br>Matthew F Traxler |
| National Science Foundation | PAPM EAGER 1650059 | Rita de Cassia Pessotti<br>Matthew F Traxler |
| Hellman Foundation | Fellow awards | Rita de Cassia Pessotti<br>Bridget L Hansen<br>Jewel N Reaso<br>Matthew F Traxler |
| U.S. Department of Energy | Genomic Science Program as part of the LLNL Biofuels SFA SCW1039 | Javier A Ceja-Navarro<br>Eoin L Brodie |

The funders had no role in study design, data collection and interpretation, or the decision to submit the work for publication.

### Author contributions

Rita de Cassia Pessotti, Conceptualization, Resources, Data curation, Formal analysis, Supervision, Validation, Investigation, Visualization, Methodology, Writing - original draft, Project administration, Writing - review and editing; Bridget L Hansen, Conceptualization, Investigation, Methodology, Writing - review and editing; Jewel N Reaso, Laila El-Hifnawi, Investigation; Javier A Ceja-Navarro, Resources, Formal analysis, Investigation, Visualization, Writing - review and editing; Eoin L Brodie, Resources, Writing - review and editing; Matthew F Traxler, Conceptualization, Supervision, Funding acquisition, Visualization, Methodology, Writing - original draft, Project administration, Writing - review and editing

### Author ORCIDs

Rita de Cassia Pessotti (iD) http://orcid.org/0000-0003-0805-8613
Bridget L Hansen (iD) https://orcid.org/0000-0001-5507-7169
Javier A Ceja-Navarro (iD) https://orcid.org/0000-0002-2954-3477
Eoin L Brodie (iD) https://orcid.org/0000-0002-8453-8435
Matthew F Traxler (iD) https://orcid.org/0000-0001-8430-595X

### Decision letter and Author response

Decision letter https://doi.org/10.7554/eLife.65091.sa1

Author response https://doi.org/10.7554/eLife.65091.sa2

## Additional files

### Supplementary files

• Supplementary file 1. Supplementary Tables S1-6. **Table S1:** Geographic location and other abiotic data of sampled beetle galleries. **Table S2:** Code and description of each microbe isolated from bessbug galleries. Antagonism assay results are represented in size of the zone of inhibition in millimeters. **Table S3:** Compounds detected in the frass of each gallery with their respective observed *m/z* of different adducts, retention time (Rt), peak height, and presence of MS2 spectrum. Compounds 1-13, 24 and 26 were annotated at level 1 identification if a MS2 spectrum of at least one adduct was detected, otherwise, they were annotated at level 2 identification. Compounds 16 and 17 were annotated at level 2 identification. Compounds 14 and 15 were annotated at level 3 identification. **Table S4:** Peak heights of compounds detected in each technical replicate of the LC-MS analyses of the environmental frass. **Table S5:** Details on the annotation of each compound. **Table S6:** GenBank accession number of each sequenced gene for microbial strains that were included in the phylogenetic study.

• Supplementary file 2. Extracted ion chromatogram (EIC, showing retention time) and MS2 spectra of each compound annotated at identification level 1 (actinomycin D, actinomycin X2, STA-21, rubiginone B2, cycloheximide, nonactin, monactin, dinactin, trinactin, tetranactin, filipin I, filipin II, filipin III, antimycin A, nocardamine, bafilomycin A1, bafilomycin B1, novobiocin, piericidin A, nigericin, beauvericin), comparing a commercial standard (top) to the culture extract of an exemplary microbe or environmental frass extract (bottom).

• Supplementary file 3. MS2 spectra of each compound annotated at identification level 2 (alteramide A, alteramide B, surugamide A), comparing a spectrum detected in the culture extract of an exemplary microbe (top) to a publicly available spectrum on the MassIVE repository (bottom). Publicly available spectra can be found at: alteramide A, alteramide B: f.MSV000079516/ccms_peak/Labelled/R5_lab_J1074_pre.mzXML; surugamide A: f.MSV000079519/ ccms_peak/Unlabelled/ A1_unlab_J1074_pre.mzXML (accessed on June 2020).

• Supplementary file 4. MS2 spectrum of each compound putatively annotated at identification level 3 (filipin IV, fungichromin), detected in the culture extract of an exemplary microbe. Annotations were made based on fragmentation similarities with other analogs of the same family annotated at identification level 1 (filipins I–III).

• Supplementary file 5. Three exemplary total ion chromatograms (TICs) of the LC-MS/MS analysis performed on environmental samples. (**A**) Pupal chamber material; (**B**, **C**) old frass, plus the extracted ion chromatogram (EIC) of an exemplary compound detected in each sample. The MS1 spectrum refers to the main peak detected on each EIC, highlighting two adducts of each compound.

• Supplementary file 6. Strains of *Streptomyces padanus* were isolated from 19/22 of the sampled *O. disjunctus* galleries. (**A**) *S. padanus* P333 growing on an ISP2-agar plate after 7 days of incubation at 30˚C. (**B**) Galleries that *S. padanus* was isolated from.

• Supplementary file 7. Average coverage distribution of *Streptomyces*-derived genes identified in the metagenome of the different gut compartments of *O. disjunctus*. The figure shows the prevalence of the *Streptomyces*-derived genes in all four gut compartments with significantly higher normalized-coverage in the posterior hindgut (PHG), the region where frass is compacted prior to its excretion. In each boxplot, a point represents a single gene per category and its detected coverage, and the diamond symbols represent the mean. The box boundaries represent the first and third quartiles of the distribution, and the median is represented as the horizontal line inside each box. Boxplots whiskers span 1.5 times the interquartile range of the distribution. FG: foregut; MG: midgut; AHG: anterior hindgut. Statistical differences were evaluated with Kruskal–Wallis test, and pairwise comparisons were done using a two-sided Wilcoxon test with p-values adjusted using the Benjamini–Hochberg method. Contigs and RPKM-normalized coverage data reported in *Ceja-Navarro et al., 2019* were used to generate this figure. Contigs were aligned against the NCBI non-

redundant database using the DIAMOND software (*Buchfink et al., 2015*) and the 'long reads' option. The obtained alignment was imported into MEGAN (*Huson et al., 2018*) and the taxonomy assigned using MEGAN's LCA algorithm for long reads. Coverage data across the four regions of *O. disjunctus*' gut was retrieved for contigs identified as taxonomically derived from *Streptomyces* sp.

- Supplementary file 8. *Metarhizium anisopliae* strains P016 and P287 phenotypes after 10 days growing on PDA plates incubated at 25℃ under constant light. Magnification: 7×.

- Transparent reporting form

## Data availability

The LC-MS/MS raw data was deposited in the Mass Spectrometry Interactive Virtual Environment database (MassIVE, https://massive.ucsd.edu/), datasets noted below. Multi-locus Sequence Analysis (MLSA) for all actinomycete strains included in our phylogenetic analysis data was supplied to Genbank under accession numbers MW073321–MW073387 and MW076236–MW076436.

The following datasets were generated:

| Author(s) | Year | Dataset title | Dataset URL | Database and Identifier |
|---|---|---|---|---|
| Rita de Cassia P | 2020 | Untargeted LC-MS/MS analysis of streptomycetes associated with Odontotaenius disjunctus (bessbug beetle) frass | https://massive.ucsd.edu/ProteoSAFe/dataset.jsp?task=8744db6238f741ceb7-f72878e54a757f | MassIVE, MSV0000 86314 |
| Rita de Cassia P | 2020 | Untargeted LC-MS/MS analysis of Odontotaenius disjunctus (bessbug beetle) frass samples | https://massive.ucsd.edu/ProteoSAFe/dataset.jsp?task=cb5dc855a811433c-ba1ea7937f6e5791 | MassIVE, MSV0000 86312 |
| Rita de Cassia P | 2020 | Untargeted LC-MS analysis of Odontotaenius disjunctus (bessbug beetle) frass samples | https://massive.ucsd.edu/ProteoSAFe/dataset.jsp?task=1363f0d204a44a9c895-c772acd7a2894 | MassIVE, MSV0000 86311 |
| Rita de Cassia P | 2020 | LC-MS/MS analysis of commercial standards | https://massive.ucsd.edu/ProteoSAFe/dataset.jsp?task=1784dac0cb254-ced86dcd399cee7dd22 | MassIVE, MSV0000 86330 |
| Rita de Cassia P | 2020 | Targeted LC-MS/MS analysis of Odontotaenius disjunctus (bessbug beetle) frass samples | https://massive.ucsd.edu/ProteoSAFe/dataset.jsp?task=e258b33e-d35e4f008b884c0e5-ca3a8f6 | MassIVE, MSV0000 86423 |

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
