## [Decision Letter]

**Acceptance summary:**

Pessotti et al. explore the chemical ecology of passalid beetles, providing a rich starting point to further dissect this model system, with broader implications for host-microbiome interactions. More broadly, this interdisciplinary study emphasizes the value of natural survey data and follow-on laboratory experiments.

**Decision letter after peer review:**

Thank you for submitting your article "Enhanced antimicrobial diversity in situ through relaxed symbiont specificity in an insect/actinomycete partnership" for consideration by *eLife*. Your article has been reviewed by three peer reviewers, including Peter Turnbaugh as the Reviewing Editor and Reviewer #1, and the evaluation has been overseen by Christian Rutz as the Senior Editor.

The reviewers have discussed their reviews with one another, and the Reviewing Editor has drafted this decision letter to help you prepare a revised submission.

Summary:

Pessotti et al. explore the chemical ecology of passalid beetles, providing a rich starting point to further dissect this model system, with broader implications for host-microbiome interactions. Overall, the manuscript is well-written and impressive in scope, providing both natural survey data and extensive follow-up experiments.

Essential revisions:

1. The claim for 'relaxed symbiont specificity' (from title and abstract) is not well supported. Actually, the work provides support for O. disjuctus having specificity for *Streptomycespadanus*. We believe that you will probably agree that it is extremely likely that some of the actinomycetes you isolated are not actually symbionts of *O. disjuctus*. In addition to beetle symbionts, you almost certainly obtained strains associated with the decomposing logs the beetles consume and live within, as well as environmental contaminants (e.g., from soil, etc). On line 139, you indicate you sampled wood from within galleries, but it does not appear to be used in a way that can inform on wood associated strains versus frass/beetle isolates. Likewise, it would have been helpful to sample decomposing parts of logs not colonized by beetles, to help determine if any of the commonly isolated *Streptomyces* are just ubiquitous within this plant material. We envision that you can address these points without doing more experiments, by improving on the analyses, such as making better use of the metagenomes you have available for O. disjuctus and including analyses of all the sequenced strains (not removing the duplicates, line 221-222) and evaluating the source of each (frass, beetles, wood, etc). Alternatively, you could just rework the manuscript to avoid making claims around specificity and be more conservative with the use of the word 'symbiont' universally for all strains.

2. You have not directly shown a benefit to the beetles from the *Streptomyces*. We recognize that experiments directly testing this would be difficult, and a lot of compelling indirect support is provided. Nevertheless, it remains possible that *Streptomyces* strains produce antimicrobials to inhibit competitors for the wood substrate/frass and that this provides no immune benefit to beetles. We think you can address this in three possible ways. First, make a stronger case for the benefit around gallery hygiene (the interesting finding of Beauvericin in a significant number of galleries in the field is something you can use to help with this). Second, you can make the case for some of these *Streptomyces* having the potential for being nutritional symbionts (see point 4 included). Third, reword the paper to reduce the sometimes very strong statements around conferred antimicrobial defense for beetles (e.g., from the end of the abstract 'Together, these findings support a model in which coprophagy as a vertical transmission mechanism leads to relaxed symbiote specificity, resulting in a rich and dynamic repertoire of antimicrobials that insulates *O. disjunctus* against the evolution of pathogen resistance.'). The linking to evolution of pathogen resistance seems a stretch too far.

3. More is needed to test the importance of coprophagy for the vertical transmission of symbionts. To test vertical transmission, you could sample the gallery, the frass, and the intestinal gut of the larvae (potentially also the pupae).

4. Isolate fungi in that ecosystem that could account for the presence of Beauvericin.

5. Additional information is needed to assess the validity of the chemical analysis reported. As it stands, it is not a metabolomic profiling but a chemical analysis to find known compounds related to microorganisms. With regards to experimental details, please include if DIA or DDA experiments were used to obtain MS/MS experiments and add three sample TICs (frass from adult, from larvae and gallery if available) in the supplementary material. We suggest performing a multivariate analysis of a feature table from the MS experiments, and validating your findings on prevalence of compounds using PLS-DA. See for example these works: Ernst, M et al. Assessing specialized metabolite diversity in the cosmopolitan plant genus Euphorbia L. Front. Plant. Sci. 2019. 10:846. And Reverter, M et al. Metabolomics and Marine Biotechnology: Coupling Metabolite Profiling and Organism Biology for Discovery of New Compounds. Front. Mar. Sci. 2020. doi.org/10.3389/fmars.2020.613471

[Editors' note: further revisions were suggested prior to acceptance, as described below.]

Thank you for submitting your article "High antimicrobial richness in situ is linked to loose partner specificity in an insect/actinomycete association" for consideration by *eLife*. Your revised article has been reviewed by three peer reviewers, including Peter Turnbaugh as the Reviewing Editor and Reviewer #1, and the evaluation has been overseen by Christian Rutz as the Senior Editor. Following our discussions, the Reviewing Editor has drafted this letter to help you prepare a revised submission.

Essential Revisions:

Please amend the title and abstract to qualify the claims of 'high antimicrobial richness' and its association with 'loose partner specificity'; for clarification, please see the detailed comments by Reviewer #2 below, which refer to essential revision points 1-3 from the first round of review.

Reviewer #2:

Point 1:

De Bary coined the term 'symbiosis' for lichens, where it is clear that the organisms involved are clearly 'living together' and not transient. The issue here is not around 'mutualism, commensalism, and parasitism', but rather determining if strains are 'living together' with the beetles or not.

The use of the word 'loose' is better than 'relaxed' in the new version of the manuscript. Nevertheless, the evidence in support of an association between *O. disjunctus* and the so-called 'loosely associated' strains of *Streptomyces* is still limited. The authors argue that this is supported by the 'high diversity of *Streptomyces*-produced compounds' present in frass in *O. disjunctus* galleries in situ and the identification of nactins in 15/22 galleries. Of the compounds identified in situ, the Actinomycins, Bafimycins, Flipins, and Angucylinones are all associated with the closely associated clades of *Streptomyces*. That just leaves the nactins as the only compound found more than a few times in situ in frass (e.g., Alteramides in 2 galleries, piericidin A in 1 gallery). I do not find this is particularly compelling for claiming 'high antimicrobial richness' associated with 'loose partner specificity'. The evidence for in vitro production is not informative, as it assumes without evidence that the strain is 'living together' with *O. disjunctus* and that the compound is produced in situ, with the latter even being contradicted by the lack of detection in situ.

Points 2 and 3:

The authors have improved their manuscript relating to point 2 (e.g. adding the potential nutritional symbiosis). The responses to each of these two points focuses a lot on the findings around *S. padanus*. For example, *S. padanus* is shown to be a superior competitor in frass. I'm fine with this, but it argues against the claim of 'loose' associations relevant to point 1. Further, the case for coprophagy and vertical transmission is largely focused on the three main clades of *Streptomyces* found with *O. disjunctus*. I should point out that finding *Streptomyces* in the guts of larvae and adults provides only minimal evidence for vertical transmission.

---

## [Author Response]

Essential revisions:1. The claim for 'relaxed symbiont specificity' (from title and abstract) is not well supported. Actually, the work provides support for O. disjuctus having specificity for Streptomyces padanus. We believe that you will probably agree that it is extremely likely that some of the actinomycetes you isolated are not actually symbionts of O. disjuctus. In addition to beetle symbionts, you almost certainly obtained strains associated with the decomposing logs the beetles consume and live within, as well as environmental contaminants (e.g., from soil, etc). On line 139, you indicate you sampled wood from within galleries, but it does not appear to be used in a way that can inform on wood associated strains versus frass/beetle isolates. Likewise, it would have been helpful to sample decomposing parts of logs not colonized by beetles, to help determine if any of the commonly isolated Streptomyces are just ubiquitous within this plant material. We envision that you can address these points without doing more experiments, by improving on the analyses, such as making better use of the metagenomes you have available for O. disjuctus and including analyses of all the sequenced strains (not removing the duplicates, line 221-222) and evaluating the source of each (frass, beetles, wood, etc). Alternatively, you could just rework the manuscript to avoid making claims around specificity and be more conservative with the use of the word 'symbiont' universally for all strains.

In our initial submission, we adopted the broad definition of symbiont coined by De Bary in 1879, which describes symbiosis as "the living together of unlike organisms" (Martin and Schwab, 2012; Saffo, 1993) and encompasses mutualism, commensalism and parasitism. Such a broad definition might require that we consider any strain isolated from within a beetle gallery to be a potential symbiont. However, we acknowledge that there are competing definitions in the literature around the term “symbiont” (which we certainly do not wish to take a stand on here). Beyond this, outside of a few specific streptomycete clades (e.g. *S. padanus*, *S. cellostaticus* and *S. scopuliridis*), we agree with the reviewers that some of the *Streptomyces* we isolated are likely transient or new immigrants in this system. In the light of the reviewers’ comments we opted to be more conservative regarding use of the word ‘symbiont’ in the revised version of this manuscript, especially in regards to strains that fall outside of clades that are likely stably-associated with *O. disjunctus* galleries. Changes have been accordingly throughout the manuscript (lines 36, 44, 75, 86, 105, 429, 438, 543-544, 573, 608).

As noted by the reviewers, we suggest that at least three *Streptomyces* clades (*S. padanus*, *S. cellostaticus* and *S. scopuliridis*) have a stable association with *O. disjunctus*, given that these clades are composed by highly related strains which were isolated from distant galleries (~1900 km apart), and from both larvae and adults. These three clades, plus the abundance of the other streptomycetes we isolated which may or may not be true symbionts, represent a higher level of potential partner diversity compared to other well-studied archetypal insect/actinomycetes associations, such as the leafcutter ants and beewolf wasps in which only one partner strongly dominates per insect species. Beyond this, we have shown that a high diversity of streptomycete-produced compounds is present in frass in *O. disjunctus* galleries in situ. Our analyses suggest that this diversity of compounds can only be accounted for by a phylogenetically diverse set of metabolically-active *Streptomyces* being present in these galleries. This *Streptomyces* diversity must extend beyond the three stably-associated clades identified here, since for example, nactins (found in 15/22 galleries) are not produced by *S. padanus*, *S. cellostaticus*, or *S. scopuliridis*. Taken together this evidence points to a system that accommodates a comparatively higher level of partner diversity and thus a correspondingly lower level of partner specificity. The terminology for describing systems with many potential symbiotic partners seems relatively poorly defined. Thus, we settled on the term “relaxed” in our prior submission. The term “relaxed” was substituted with “loose” in the new version in an attempt to more accurately capture our meaning.

As suggested by the reviewers, we built an additional tree (using only the 16S rRNA gene) that included duplicate strains that were removed from the main tree presented in Figure 4 (due to our not wanting to saturate the tree with sequences potentially representing co-isolated clones). As perhaps the reviewers hypothesized, this new tree further reinforces the conclusions made above (i.e. a few clades are robustly associated with *O. disjunctus* frass, while others seem more ambiguous). This new tree was incorporated into the manuscript as Figure 4—figure supplement 6.

We would also like to point out a new analysis we performed in response to the feedback we received from the reviewers. We built another phylogenetic tree (using the 16S rRNA gene) with our isolates plus sequences available in GenBank for 101 *Streptomyces* from soil sampled from diverse sites around the world in a single study (Schlatter and Kinkel, 2014), and more than 200 *Streptomyces* strains isolated from diverse groups of insects including tropical passalid beetles, termites, bees, wasps and ants (Benndorf et al., 2018; Matarrita-Carranza et al., 2017; Vargas-Asensio et al., 2014). This tree showed that the majority of our isolates grouped in clades with strains isolated from diverse insects from distant parts of the world. In contrast, most of our isolates were not found in interdigitated with isolates from soil. Such a phylogenetic distribution might indicate that most of our isolates belong to lineages that form ready associations with insects. We appreciate that more work would need to be done to further explore these patterns. This new tree was incorporated into the manuscript as Figure 4—figure supplement 8.

2. You have not directly shown a benefit to the beetles from the Streptomyces. We recognize that experiments directly testing this would be difficult, and a lot of compelling indirect support is provided. Nevertheless, it remains possible that Streptomyces strains produce antimicrobials to inhibit competitors for the wood substrate/frass and that this provides no immune benefit to beetles. We think you can address this in three possible ways. First, make a stronger case for the benefit around gallery hygiene (the interesting finding of Beauvericin in a significant number of galleries in the field is something you can use to help with this). Second, you can make the case for some of these Streptomyces having the potential for being nutritional symbionts (see point 4 included). Third, reword the paper to reduce the sometimes very strong statements around conferred antimicrobial defense for beetles (e.g., from the end of the abstract 'Together, these findings support a model in which coprophagy as a vertical transmission mechanism leads to relaxed symbiote specificity, resulting in a rich and dynamic repertoire of antimicrobials that insulates O. disjunctus against the evolution of pathogen resistance.'). The linking to evolution of pathogen resistance seems a stretch too far.

This is an excellent point, with which we fundamentally agree. Any benefit the beetles derive from the panoply of antimicrobial compounds made by these *Streptomyces* likely comes from enhanced gallery hygiene, and possibly enhanced hygiene of pupal chambers. These molecules are not a substitute for the immune system of the beetles. We have made changes throughout the manuscript, as suggested, to clarify this point (lines 45-46, 106-108, 430-431).

We also agree with the reviewers that it is very likely that the antimicrobials that *Streptomyces* strains produce might be used in competition among them. A number of our experimental results and statements made in the manuscript are consistent with this idea. We tested this competition hypothesis in our pairwise frass assay between *Streptomyces*, and found that the selected strains compete strongly when growing on frass, and that *S. padanus* is a superior competitor. We also note this idea on lines 541-545: “Thus, we suggest that in the case of *O. disjunctus*, coprophagy as a microbial transmission mechanism may lead to loose partner specificity and increased competition among the streptomycete community, resulting in a correspondingly wide profile of antimicrobials with varied mechanisms of action.”. This chemical warfare may certainly have the indirect effect of keeping the galleries free of fungal pathogens. In addition, the *Streptomyces* and fungal pathogens may compete directly in the frass environment. This idea is consistent with the results shown in Figure 6D, in which all three *Streptomyces* strains directly inhibited *Metarhizium anisopliae* growth in frass.

We thank the reviewers for calling attention to the idea of nutritional symbiosis and agree that this is an important hypothesis to be discussed, even though we did not perform any specific experiments to test it. Indeed, *Streptomyces* spp. are known producers of enzymes that degrade wood components, e.g. cellulose, lignin and xylose (Book et al., 2014, 2016). Therefore, the streptomycete community associated with *O. disjunctus* might be playing a nutritional role in this system as well. In fact, work that was performed in Costa Rica on this topic analyzed the streptomycete community associated with tropical passalid beetles, which provided strong evidence that *Streptomyces* play an important role as nutritional symbionts (Vargas-Asensio et al., 2014). Consistent with this idea, when we cultured three strains directly on frass material (which is largely composed by partially digested wood components) as the only nutrient source, all strains displayed robust growth (Figure 6, Figure 6—figure supplement 1). This result indicates that at least these three *Streptomyces* could be part of the “external rumen” that provides valuable nutrients to both adults and larvae. We modified the manuscript accordingly to include the discussion of this important topic (lines 481-486).

3. More is needed to test the importance of coprophagy for the vertical transmission of symbionts. To test vertical transmission, you could sample the gallery, the frass, and the intestinal gut of the larvae (potentially also the pupae).

As presented in the grey-scale heatmap in the interior of Figure 4, we analyzed strains isolated from old frass/gallery wood (representing strains that are present within the galleries); pupal chamber material/pupal swabs (representing strains associated with pupa); and fresh frass collected inside Petri dishes from both individual adults and larvae (representing a snapshot of the intestinal gut microbial composition of these two developmental stages). We want to note that finding galleries with larvae present was challenging, and we were fortunate that we found three of them (2-DC, 15-FL and 17-LA). Even though the number of samples from the larval stage was very low (13.6% of the galleries), we isolated multiple strains from larval frass, including members of the three clades we highlighted as stable members in this system (*S. padanus, S. cellostaticus*, and *S. scopuliridis*) (Figure 4, Figure 4—figure supplement 6). These findings show that at least these three main clades are present during different developmental stages and are likely widespread within galleries across the eastern US.

Collectively, our findings support the idea that some clades are being transmitted across *O. disjunctus* generations. Given that no mycangia have been described in passalid beetles, we propose coprophagy as a possible mode for vertical transmission to the progeny, as seen for other insects (Onchuru et al., 2018). We base this on the fact that (1) frass is the major, if not the only, food source provided to the larvae by their parents, (2) strains of the three highlighted clades were isolated from adult and larval fresh frass, which is a snapshot of the gut content, and (3) the finding of *Streptomyces* DNA in the metagenomics analysis of *O. disjunctus* gut by members of our team (Appendix 1 – Figure 6). Additionally, a recent cultured-based analysis of tropical passalid beetles guts showed the presence of *Streptomyces*, indicating that this group of bacteria is present in the guts of the tropical relatives of the beetles studied here (Vargas-Asensio et al., 2014). We have revised the passages about this topic for clarity (lines 219-290, 525-549). Nonetheless, we agree that more experimentation could further substantiate the idea that vertical transmission through coprophagy occurs in this system, and we acknowledge this in the revised version (lines 547-549).

4. Isolate fungi in that ecosystem that could account for the presence of Beauvericin.

We are not exactly sure what the reviewers have in mind regarding this request. Our original intent was to assess the *O. disjunctus* system for potential actinomycete partnerships. However, the finding of beauvericin in frass from many galleries, and the presence of *Metarhizium* spp. Isolated from a gallery and from an *O. disjunctus* carcass found in the field, underscored the potential importance of fungal pathogens in this system. While we do not currently have any fungal isolates that produce beauvericin, we note that this compound identification was validated using a commercial standard (Appendix 1 – Figure 1) with high confidence. Beauvericin is a well-known fungal metabolite that has insecticidal, antimicrobial, nematocidal, and anti-tumor activity (Wang and Xu, 2012), and its origins as a mycotoxin are undisputed. For these reasons, we suggest that its detection indicates that entomopathogens like *Beauveria* spp. and/or *Fusarium* spp., which are well-documented producers of beauvericin (Hamill et al., 1969; Logrieco et al., 1998), might be present and posing a threat to the galleries. To clarify this, we made minor modifications in lines 175-176, 179-180, 473-474 and 522.

5. Additional information is needed to assess the validity of the chemical analysis reported. As it stands, it is not a metabolomic profiling but a chemical analysis to find known compounds related to microorganisms. With regards to experimental details, please include if DIA or DDA experiments were used to obtain MS/MS experiments and add three sample TICs (frass from adult, from larvae and gallery if available) in the supplementary material. We suggest performing a multivariate analysis of a feature table from the MS experiments, and validating your findings on prevalence of compounds using PLS-DA. See for example these works: Ernst, M et al. Assessing specialized metabolite diversity in the cosmopolitan plant genus Euphorbia L. Front. Plant. Sci. 2019. 10:846. And Reverter, M et al. Metabolomics and Marine Biotechnology: Coupling Metabolite Profiling and Organism Biology for Discovery of New Compounds. Front. Mar. Sci. 2020. doi.org/10.3389/fmars.2020.613471

Thank you for this comment. We welcome the opportunity to fully clarify any questions regarding our chemical analysis. All data were acquired using data-dependent analysis (DDA), sending for fragmentation the 5 most intense ions (also known as a Top5 method), excluding repetitive ions for 5 seconds (dynamic exclusion). This information was added in lines 705-706.

In terms of environmental sampling, we performed direct extractions of gallery ‘old frass’ and pupal chamber material. We did not perform extractions of fresh frass from adult/larvae, rather we reserved that material for microbial isolations. Therefore, we have added to the supplementary material the TIC of old frass extract from two randomly selected galleries and pupal chamber material extract from one randomly selected gallery, and we also provided the EIC of selected specialized metabolites detected in these samples (Appendix 1 – Figure 4).

With regard to additional information to assess the validity of the chemical analyses, we would like to point to the extensive supplementary tables that were curated by directly checking each one of the original unconverted spectra files to assess the confidence of the ions detected in the feature table: (1) Supplementary File 1C – Table S3: Identified compounds detected in the frass of each gallery with their respective observed *m/z* of different adducts, retention time (Rt), peak height, and presence of a manually checked MS2 spectrum; (2) Supplementary File 1D – Table S4: Peak heights of annotated compounds detected in each technical replicate of the LC-MS analyses of the environmental frass; (3) Supplementary File 1E – Table S5: Details on the annotation of each one of the 25 compounds. These tables are the result of a manual check of the extracted ion chromatogram (EIC) of each annotated ion. We would also like to point out to the representative MS2 spectra of annotated compounds and commercial chemical standards in Appendix 1 – Figure 1.

We agree that multivariate analyses are very useful for comparing and detecting patterns in untargeted metabolomics data. To this end we have performed an additional hierarchical clustering analysis (HCA) on the feature table from the untargeted runs of the extracts from each of our *Streptomyces* isolates grown in vitro, and compared it to the phylogenetic tree using a tanglegram (Figure 4—figure supplement 2). This analysis was inspired by the excellent Ernst *et al.* 2019 paper pointed out by the reviewers. In this case we see a very clear correlation between the phylogenetic relationships of the isolates and the content of their metabolomes.

Near the beginning of this project, we performed multiple NMDS analyses of the frass extracts from all the galleries, but we were not confident of any patterns related to latitude, longitude, altitude, north/central/south subdivision, or frass pH. Our current best guess is that the chemical features present in the frass extracts are probably dominated by compounds associated with the various wood types. *O. disjunctus* does not feed on logs of a specific tree species, and we did not control for this when sampling galleries. We have not performed DNA analyses to assess the tree species of each gallery. Thus, we elected not to include these analyses, since they could potentially invite misleading interpretations. Instead, we chose to pursue a more conservative analysis of the frass, presenting only presence/absence data of the known microbial metabolites that we could annotate with confidence, and stepping away from a broader metabolomic profiling of this material. We made minor changes in the text to make it clear that we focused on a small subset of compounds instead of full metabolomic profiling of the frass material (lines 98 and 157).

References:

Benndorf, R., Guo, H., Sommerwerk, E., Weigel, C., Garcia-Altares, M., Martin, K., Hu, H., Küfner, M., de Beer, Z. W., Poulsen, M., and Beemelmanns, C. (2018). Natural products from actinobacteria associated with fungus-growing termites. Antibiotics, 7(3), 83. https://doi.org/10.3390/antibiotics7030083Blaženović, I., Kind, T., Ji, J., and Fiehn, O. (2018). Software tools and approaches for compound identification of LC-MS/MS data in metabolomics. Metabolites, 8(31). https://doi.org/10.3390/metabo8020031Book, A. J., Lewin, G. R., McDonald, B. R., Takasuka, T. E., Doering, D. T., Adams, A. S., Blodgett, J. A. V., Clardy, J., Raffa, K. F., Fox, B. G., and Currie, C. R. (2014). Cellulolytic Streptomyces strains associated with herbivorous insects share a phylogenetically linked capacity to degrade lignocellulose. Applied and Environmental Microbiology, 80(15), 4692–4701. https://doi.org/10.1128/AEM.01133-14Book, A. J., Lewin, G. R., McDonald, B. R., Takasuka, T. E., Wendt-Pienkowski, E., Doering, D. T., Suh, S., Raffa, K. F., Fox, B. G., and Currie, C. R. (2016). Evolution of High Cellulolytic Activity in Symbiotic Streptomyces through Selection of Expanded Gene Content and Coordinated Gene Expression. PLoS Biology, 14(6), e1002475. https://doi.org/10.1371/journal.pbio.1002475Hamill, R. L., Higgens, C. E., Boaz, H. E., and Gorman, M. (1969). The structure of beauvericin, a new depsipeptide antibiotic toxic to artemia salina. Tetrahedron Letters, 10(49), 4255–4258. https://doi.org/10.1016/S0040-4039(01)88668-8Hollstein, U. (1974). Actinomycin. Chemistry and mechanism of action. Chemical Reviews, 74(6), 625–652. https://doi.org/10.1021/cr60292a002Logrieco, A., Moretti, A., Castella, G., Kostecki, M., Golinski, P., Ritieni, A.,and Chelkowski, J. (1998). Beauvericin production by Fusarium species. Applied and Environmental Microbiology, 64(8), 3084–3088. https://doi.org/10.1128/aem.64.8.3084-3088.1998Martin, B. D., and Schwab, E. (2012). Current Usage of Symbiosis and Associated Terminology. International Journal of Biology, 5(1), 32–45. https://doi.org/10.5539/ijb.v5n1p32Matarrita-Carranza, B., Moreira-Soto, R. D., Murillo-Cruz, C., Mora, M., Currie, C. R., and Pinto-Tomas, A. A. (2017). Evidence for widespread associations between neotropical hymenopteran insects and Actinobacteria. Frontiers in Microbiology, 8, 2016. https://doi.org/10.3389/fmicb.2017.02016Onchuru, T. O., Javier Martinez, A., Ingham, C. S., and Kaltenpoth, M. (2018). Transmission of mutualistic bacteria in social and gregarious insects. Current Opinion in Insect Science, 28, 50–58. https://doi.org/10.1016/j.cois.2018.05.002Pressman, B. C. (1976). Biological applications of ionophores. Annual Review of Biochemistry, 45, 501–530. https://doi.org/10.1146/annurev.bi.45.070176.002441Saffo, M. B. (1993). Coming to Terms with a Field: Words and Concepts in Symbiosis. Symbiosis, 14, 17–31.Schlatter, D. C., and Kinkel, L. L. (2014). Global biogeography of Streptomyces antibiotic inhibition, resistance, and resource use. FEMS Microbiology Ecology, 88(2), 386–397. https://doi.org/10.1111/1574-6941.12307Song, H., Wang, R., Wang, S., and Lin, J. (2005). A low-molecular-weight compound discovered through virtual database screening inhibits Stat3 function in breast cancer cells. Proceedings of the National Academy of Sciences of the United States of America, 102(13), 4700–4705. https://doi.org/10.1073/pnas.0409894102Sumner, L. W., Amberg, A., Barrett, D., Beale, M. H., Beger, R., Daykin, C. A., Fan, T. W. M., Fiehn, O., Goodacre, R., Griffin, J. L., Hankemeier, T., Hardy, N., Harnly, J., Higashi, R., Kopka, J., Lane, A. N., Lindon, J. C., Marriott, P., Nicholls, A. W., … Viant, M. R. (2007). Proposed minimum reporting standards for chemical analysis: Chemical Analysis Working Group (CAWG) Metabolomics Standards Initiative (MSI). Metabolomics, 3(3), 211–221. https://doi.org/10.1007/s11306-007-0082-2Tang, J., Wennerberg, K., and Aittokallio, T. (2015). What is synergy? The Saariselka agreement revisited. Frontiers in Pharmacology, 6, 181. https://doi.org/10.3389/fphar.2015.00181Vargas-Asensio, G., Pinto-Tomas, A., Rivera, B., Hernandez, M., Hernandez, C., Soto-Montero, S., Murillo, C., Sherman, D. H., and Tamayo-Castillo, G. (2014). Uncovering the cultivable microbial diversity of costa rican beetles and its ability to break down plant cell wall components. PLoS ONE, 9(11), e113303. https://doi.org/10.1371/journal.pone.0113303Wang, Q., and Xu, L. (2012). Beauvericin, a bioactive compound produced by fungi: A short review. Molecules, 17, 2367–2377. https://doi.org/10.3390/molecules17032367

[Editors' note: further revisions were suggested prior to acceptance, as described below.]

Essential Revisions:Please amend the title and abstract to qualify the claims of 'high antimicrobial richness' and its association with 'loose partner specificity'; for clarification, please see the detailed comments by Reviewer #2 below, which refer to essential revision points 1-3 from the first round of review.Reviewer #2:Point 1:De Bary coined the term 'symbiosis' for lichens, where it is clear that the organisms involved are clearly 'living together' and not transient. The issue here is not around 'mutualism, commensalism, and parasitism', but rather determining if strains are 'living together' with the beetles or not.The use of the word 'loose' is better than 'relaxed' in the new version of the manuscript. Nevertheless, the evidence in support of an association between O. disjunctus and the so-called 'loosely associated' strains of Streptomyces is still limited. The authors argue that this is supported by the 'high diversity of Streptomyces-produced compounds' present in frass in O. disjunctus galleries in situ and the identification of nactins in 15/22 galleries. Of the compounds identified in situ, the Actinomycins, Bafimycins, Flipins, and Angucylinones are all associated with the closely associated clades of Streptomyces. That just leaves the nactins as the only compound found more than a few times in situ in frass (e.g., Alteramides in 2 galleries, piericidin A in 1 gallery). I do not find this is particularly compelling for claiming 'high antimicrobial richness' associated with 'loose partner specificity'. The evidence for in vitro production is not informative, as it assumes without evidence that the strain is 'living together' with O. disjunctus and that the compound is produced in situ, with the latter even being contradicted by the lack of detection in situ.Points 2 and 3:The authors have improved their manuscript relating to point 2 (e.g. adding the potential nutritional symbiosis). The responses to each of these two points focuses a lot on the findings around S. padanus. For example, S. padanus is shown to be a superior competitor in frass. I'm fine with this, but it argues against the claim of 'loose' associations relevant to point 1. Further, the case for coprophagy and vertical transmission is largely focused on the three main clades of Streptomyces found with O. disjunctus. I should point out that finding Streptomyces in the guts of larvae and adults provides only minimal evidence for vertical transmission.

We were encouraged by last round of review, in which you requested changes be made to the title and abstract. We have made the requested revisions, which I summarize below:

1. Revised the title from “High antimicrobial richness in situ is linked to loose partner specificity in an insect/actinomycete association” to “Multiple lineages of *Streptomyces* produce antimicrobials within passalid beetle galleries across eastern North America”

2. Revised the abstract to be in line with the requests of reviewer, ie. re-phrasing our interpretation to qualify the claims around 'high antimicrobial richness' and its association with 'loose partner specificity'.

3. Revised the Impact Statement to be in line with title and abstract.

4. Revised a several lines of text in the paper (L105-107, L535-540, and L605) to also align with the title and abstract.